# CELF RNA binding proteins promote axon regeneration in *C. elegans* and mammals through alternative splicing of Syntaxins

Lizhen Chen[1,2†], Zhijie Liu[3], Bing Zhou[4], Chaoliang Wei[4], Yu Zhou[4], Michael G Rosenfeld[2,3], Xiang-Dong Fu[4], Andrew D Chisholm[1*], Yishi Jin[1,2,4*]

[1]Section of Neurobiology, University of California, San Diego, Division of Biological Sciences, San Diego, United States; [2]Howard Hughes Medical Institute, University of California, San Diego, United States; [3]Department of Medicine, University of California, San Diego, School of Medicine, San Diego, United States; [4]Department of Cellular and Molecular Medicine, University of California, San Diego, School of Medicine, San Diego, United States

**Abstract** Axon injury triggers dramatic changes in gene expression. While transcriptional regulation of injury-induced gene expression is widely studied, less is known about the roles of RNA binding proteins (RBPs) in post-transcriptional regulation during axon regeneration. In *C. elegans* the CELF (CUGBP and Etr-3 Like Factor) family RBP UNC-75 is required for axon regeneration. Using crosslinking immunoprecipitation coupled with deep sequencing (CLIP-seq) we identify a set of genes involved in synaptic transmission as mRNA targets of UNC-75. In particular, we show that UNC-75 regulates alternative splicing of two mRNA isoforms of the SNARE Syntaxin/*unc-64*. In *C. elegans* mutants lacking *unc-75* or its targets, regenerating axons form growth cones, yet are deficient in extension. Extending these findings to mammalian axon regeneration, we show that mouse *Celf2* expression is upregulated after peripheral nerve injury and that *Celf2* mutant mice are defective in axon regeneration. Further, mRNAs for several Syntaxins show CELF2 dependent regulation. Our data delineate a post-transcriptional regulatory pathway with a conserved role in regenerative axon extension.

**\*For correspondence:** chisholm@ucsd.edu (ADC); yijin@ucsd.edu (YJ)

**Present address:** †Department of Cellular and Structural Biology, University of Texas Health Science Center at San Antonio, Barshop Institute for Longevity and Aging Studies, San Antonio, United States

**Competing interests:** The authors declare that no competing interests exist.

## Introduction

Axon regeneration requires coordinated gene expression at many levels (*Benowitz et al., 1981*; *Gervasi et al., 2003*; *Glasgow et al., 1992*; *Skene and Willard, 1981*). While much work has focused on injury-regulated gene transcription, increasing evidence points to roles for post-transcriptional regulation of mRNAs by RNA binding proteins (RBPs). In rodents, the Zipcode Binding Protein ZBP1 can bind axonal mRNAs and affect peripheral nerve regeneration via mRNA transport and decay (*Donnelly et al., 2011*). In Xenopus, hnRNP K binds mRNAs of growth-associated proteins such as GAP43 and NF-M and promotes protein synthesis in optic nerve regeneration (*Liu et al., 2012*). Recently, the conserved RNA 3'-terminal phosphate cyclase has been identified as an inhibitor of axon regeneration in *C. elegans,* Drosophila and mouse, acting through RNA repair and splicing (*Kosmaczewski et al., 2015*; *Song et al., 2015*). Despite these advances, mechanistic understanding of the roles of RBPs in axon regeneration remains limited.

CELF (CUG-BP and ETR-3-like Factor) family RNA binding proteins are highly conserved throughout animals (*Dasgupta and Ladd, 2012*). All six mammalian CELF proteins are expressed in the nervous system and several have been implicated in neuronal alternative splicing (*Ladd, 2013*). Analysis of *Celf* mutant mice has begun to reveal their roles in neuronal development and behavior

**eLife digest**  Nerve cells or neurons carry information around the body along projections known as axons. An injury or trauma, such as a stroke, can damage the axons and lead to permanent disability because the damaged axons fail to regenerate over long distances.

Axon damage triggers large changes in the activity of many genes that promote regeneration. When a gene is active, its DNA is copied to make molecules of messenger RNA (mRNA), which are then used as templates to make proteins. Many mRNAs undergo a process called alternative splicing, in which different combinations of mRNA sections may be removed from the final molecule. This enables a single gene to produce more than one type of protein.

Recent studies point to an important role for so-called RNA binding proteins in regulating the alternative splicing process. An RNA binding protein called UNC-75 in a worm known as *Caenorhabditis elegans* has previously been shown to be involved in axon regeneration, but it was not clear how UNC-75 acts on neurons. Here, Chen et al. combined a technique called CLIP-seq (Cross-linking ImmunoPrecipitation-deep sequencing) with genetic testing to identify the mRNAs that UNC-75 regulates during axon regeneration.

The experiments found a set of *C. elegans* genes required for information to pass between neurons whose mRNAs are also targeted by UNC-75. Many of these genes are also required for axon regeneration. Chen et al. studied one of the mRNA targets – which encodes a protein called syntaxin – in more detail and found that the syntaxin mRNA is required for regenerating axons over long distances. UNC-75 alternatively splices this mRNA to produce a particular form of syntaxin that is mainly found in neurons. Mutant worms that lack either UNC-75 or syntaxin are unable to properly regenerate axons over long distances.

Further experiments show that a mouse protein known as CELF2 that is equivalent to worm UNC-75 plays a similar role in regenerating axons. Moreover, mouse CELF2 restores the ability of worm neurons that lack UNC-75 to regenerate. Like worm UNC-75, the mouse protein is also involved in alternative splicing of syntaxin. The next step is to examine the other mRNA targets of UNC-75 to find out what role they play in axon regeneration and other processes in neurons.

(*Dev et al., 2007*; *Dougherty et al., 2013*; *Kress et al., 2007*; *Wagnon et al., 2012*; *Yang et al., 2007*). *Celf4* deficient mice exhibit a seizure disorder (*Wagnon et al., 2012*; *Yang et al., 2007*), whereas *Celf6* mutant mice display abnormal behaviors and reduced brain serotonin (*Dougherty et al., 2013*). However, CELF proteins have not previously been examined in the context of axon regeneration.

Here, we addressed the roles of CELF proteins in axon regeneration, focusing on *C. elegans* UNC-75 and mouse CELF2, both of which are localized to the nucleus (*Loria et al., 2003*; *Otsuka et al., 2009*). To identify direct targets of UNC-75 in *C. elegans* neurons we performed neuronal CLIP-seq to locate UNC-75 binding sites. Many UNC-75 target sites are in genes involved in synaptic transmission. We show that UNC-75 binding to an intronic site of UNC-64/Syntaxin promotes expression of neuronal UNC-64/Syntaxin isoforms. Loss of UNC-75 or of UNC-64 causes distinctive phenotypes in which regenerative growth cones are formed but are unable to extend. Overexpression of UNC-64 in *unc-75* null mutants can rescue axon regeneration defects, indicating that UNC-64 is a major target of UNC-75 in regenerating neurons. Extending these findings to mammals, we find that mouse *Celf2* expression is induced by axon injury and that CELF2 is required for effective peripheral axon regeneration. Furthermore, we identify multiple *Syntaxin* genes as CELF2 targets. Together, our data reveal a conserved pro-regeneration pathway operating at the level of alternative splicing.

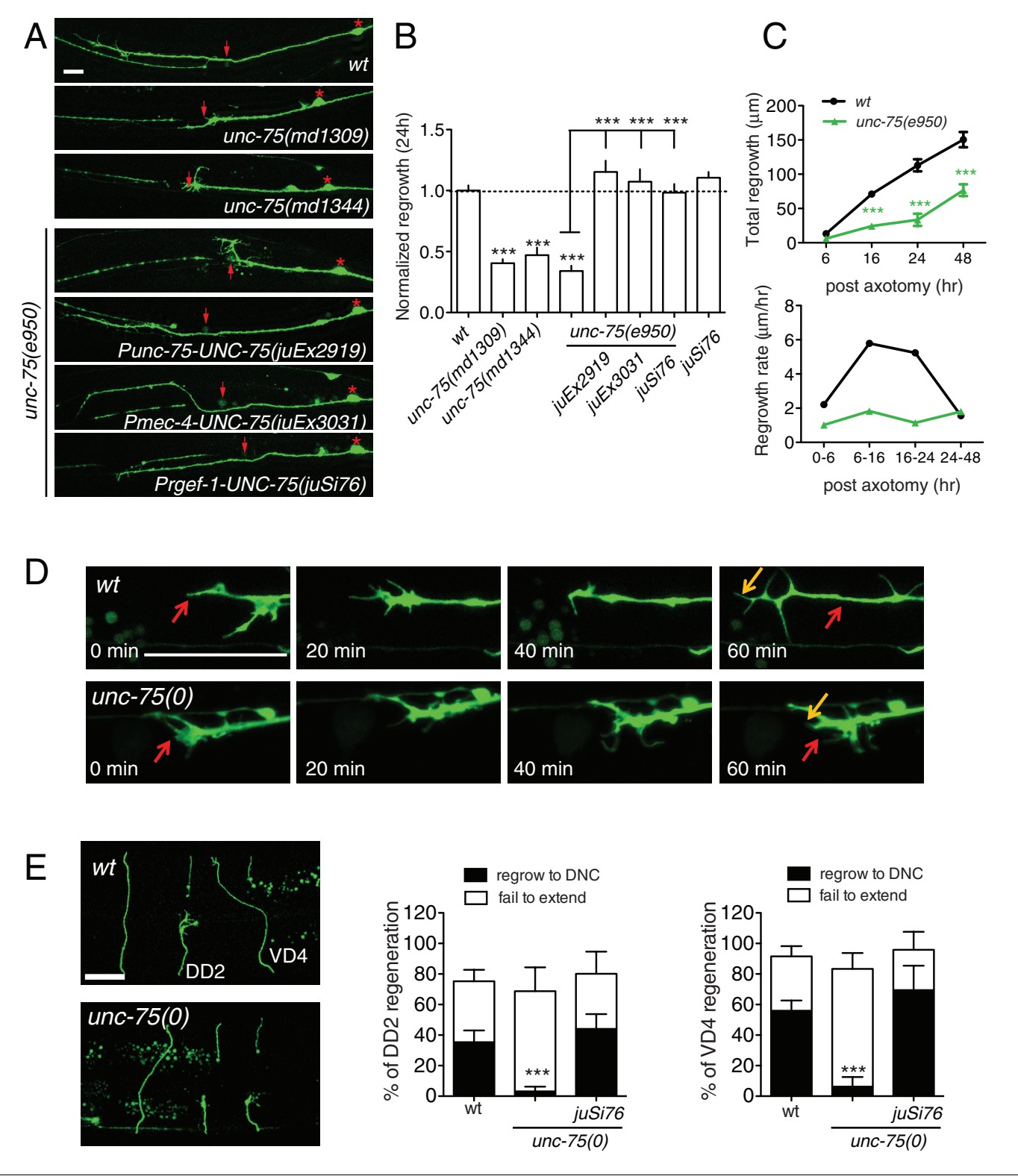

**Figure 1.** UNC-75 is required cell autonomously for axon regeneration. (A) Images of regenerating PLM axons at 24 hr post axotomy; anterior is to the left and dorsal up. Red asterisk: PLM cell body; red arrow: injury site. (B) Quantitation of PLM axon regrowth 24 hr post laser axotomy, normalized to wild type. The *unc-75* alleles *md1309* and *md1344* display defects in axon regrowth similar to *e950*. The *unc-75(e950)* PLM regrowth defect is rescued by multicopy and single copy *unc-75(+)* transgenes. Statistics: One-way ANOVA with Bonferroni post test. (C) *unc-75(e950)* animals showed reduced axon regrowth at all time points examined, and reduced growth rate 0–24 hr post axotomy. (D) Representative image series from time-lapse movies of

*Figure 1 continued on next page*

*Figure 1 continued*

the tip of a regenerating PLM axon from wild-type and *unc-75(e950)* animals starting at 14 hr post axotomy; see *Video 1* and *2*. Red and orange arrows point to the ends of regenerating axons at 14 and 15h post axotomy respectively. (**E**) *unc-75(e950)* is defective in regeneration of GABAergic motor neurons [marked by P*unc-25*-GFP(*juIs76*)]; this was rescued by P*rgef-1*-UNC-75 (*juSi76*). Images of GABAergic motor neuron commissures at 24 hr post axotomy. DD2 and VD4 were axotomized, and VD3 was uncut. Red arrowheads indicate the ends of regenerating or non-regenerating axons. Scale: 10 μm. Bar charts showing reduced regrowth of DD2 and VD4 neurons; N = 30–52.

The following figure supplements are available for figure 1:

**Figure supplement 1.** *unc-75* alleles and role of nuclear localization.

**Figure supplement 2.** Regenerative growth cones in *unc-75* and other mutants.

**Figure supplement 3.** *unc-75* acts in parallel to other axon regeneration pathways.

## Results

### UNC-75/CELF functions cell autonomously in sensory and motor axon regeneration

PLM axon development was normal in all *unc-75* mutants tested except that around 5% of PLMs lacked ventral synaptic branches. PLM axon regeneration in *unc-75* mutants was reduced to 30–40% of wild type levels (*Figure 1A,B*). Interestingly, *unc-75(md1344),* a small deletion affecting the last exon encoding part of RRM3 and a nuclear localization signal (*Figure 1—figure supplement 1A*), displayed impairment in regrowth equivalent to the null mutants (*Figure 1A,B*). Transgenic animals expressing *Pmec-4-GFP::UNC-75ΔNLS*(aa 1–472) showed diffuse fluorescence throughout the cell, as compared to full-length GFP::UNC-75, which localizes to neuronal nuclei (*Figure 1—figure supplement 1B*) (*Loria et al., 2003*), suggesting that nuclear localization may be critical for UNC-75 function in axon regeneration. We further tested whether *unc-75* affected regeneration of GABAergic motor neurons. In wild type, around 35% of DD2 commissures and 50% of VD4 commissures regrow to the dorsal cord by 24 hr after axotomy, whereas in *unc-75* mutants fewer than 10% of commissures regrew to the dorsal cord (*Figure 1E*). Thus UNC-75 is critical for regenerative regrowth of multiple neuron types.

We rescued *unc-75* PLM and motor axon regrowth defects with transgenes expressing UNC-75 under its own promoter, or with a single copy transgene expressing UNC-75 under the control of the pan-neuronal *rgef-1* promoter (*Figure 1A,B*). Moreover, expression of UNC-75 using a touch neuron specific promoter (*Pmec-4*) fully rescued PLM regrowth, indicating that UNC-75 acts cell-autonomously (*Figure 1A,B*). Expression of UNC-75 in wild type animals did not enhance regrowth, indicating that UNC-75 is necessary but not sufficient to promote axon regeneration (*Figure 1B*). A mutant *unc-75* cDNA lacking the NLS, expressed under its own promoter, failed to rescue behavioral defects of *unc-75* mutant (*Loria et al., 2003*) and did not significantly rescue the regeneration defects (*Figure 1—figure supplement 1C*), supporting its function in the nucleus. In *unc-75* mutant neurons the rate of axon extension was reduced at multiple time points after injury (*Figure 1C*), despite the presence of growth cones at the tip of the regrowing axons (*Figure 1A*). Using time-lapse imaging we found that wild-type PLM regenerative growth cones tended to be small and transient, and converted rapidly to elongating axons. In contrast, *unc-75* mutants frequently displayed

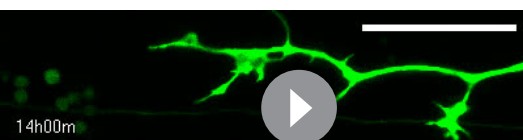

**Video 1.** Time-lapse movie of the tip of a regenerating PLM axon from a wild type animal, starting at 14 hr post axotomy, ending at 15 hr post axotomy.

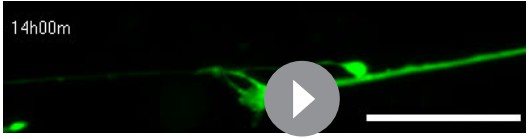

**Video 2.** Time-lapse movie of the tip of a regenerating PLM axon from an *unc-75(e950)* mutant, starting at 14 hr post axotomy, ending at 15 hr post axotomy.

regenerative growth cones with dynamic filopodial protrusions, but these did not drive long-range axon extension (*Figure 1—figure supplement 1D* and *Video 1,2*). Indeed, we observed growth cones more frequently in non-regenerating PLM axons in *unc-75* mutants than in the wild type (*Figure 1—figure supplement 2*), consistent with previous observations that presence of a growth cone does not correlate highly with axon extension (*Chen et al., 2011*; *Edwards and Hammarlund, 2014*). Injured motor axons in *unc-75* mutants also formed regenerative growth cones that failed to extend (*Figure 1E*). Thus UNC-75 is required for regenerative axon extension but is either dispensable or a negative regulator of regenerative growth cone formation.

The *C. elegans* Elav-like RNA binding protein EXC-7 functions partially redundantly with UNC-75 in synaptic transmission (*Loria et al., 2003*), and co-regulates overlapping mRNA splicing events (*Norris et al., 2014*). *exc-7* mutants displayed normal PLM axon regeneration, and *unc-75 exc-7* double mutants resembled *unc-75* single mutants in regeneration (*Figure 1—figure supplement 3*). *fox-1*/RBFOX, another known interactor of *unc-75* in neuronal alternative splicing (*Kuroyanagi et al., 2013a*), is also not required for PLM regrowth (*Chen et al., 2011*). These observations suggest UNC-75 functions in axon regeneration non-redundantly with these known neuronal splicing regulators.

We next addressed how *unc-75* interacted with the regrowth promoting MAP kinase kinase kinase DLK-1 and the regrowth inhibiting factor EFA-6. In *C. elegans*, PLM and motor neuron axon regeneration is completely blocked in null mutants of *dlk-1* and is increased in animals overexpressing active DLK-1 (*Yan and Jin, 2012*). Double mutants *unc-75(0); dlk-1(0)* did not further reduce PLM axon regrowth compared to *dlk-1(0),* and the regrowing axons had no regenerative growth cones, resembling *dlk-1(0)* (*Figure 1—figure supplement 2A,3B*). Overexpression of DLK-1 in *unc-75(0)* partially improved regrowth compared to *unc-75(0)* single mutants (*Figure 1—figure supplement 3C*), suggesting that *unc-75* and *dlk-1* likely function in parallel. The conserved protein EFA-6 regulates axonal microtubule dynamics, and loss of EFA-6 strongly enhances PLM axon regrowth (*Chen et al., 2011*; *2015*). *efa-6(0)* partially rescued *unc-75(0)* PLM regrowth defects (*Figure 1—figure supplement 3C*), but not the growth cone phenotype (*Figure 1—figure supplement 2*), suggesting that EFA-6 and UNC-75 function in parallel. Moreover, *dlk-1* and *efa-6* transcript levels were normal in *unc-75* mutants (*Figure 1—figure supplement 3D*). Thus, these analyses suggest the UNC-75 axon extension pathway acts partly independently of the DLK-1 and EFA-6 axon regeneration regulators.

## UNC-75 mRNA targets are enriched for genes involved in RNA binding and synaptic transmission

To dissect how UNC-75 regulates axon regeneration, we isolated RNAs bound by UNC-75 in neurons (*Figure 2A*). We expressed functional FLAG-tagged UNC-75 in neurons in *unc-75(0)* mutants, in which the locomotion defect was rescued by the FLAG::UNC-75 transgene (*Figure 1—figure supplement 1D*). We then performed <u>c</u>ross<u>l</u>inking <u>i</u>mmuno<u>p</u>recipitation coupled with deep <u>seq</u>uencing (CLIP-seq) (*Figure 2—figure supplement 1A*) (see Materials and methods). We used two methods to map the unique reads onto *C. elegans* genome (*Figure 2A*), and also manually inspected the genomic loci containing UNC-75 CLIP peak positions. We identified 533 functionally annotated protein-coding genes as UNC-75 targets (*Supplementary file 1*). 79% of the peaks in protein-coding genes were in intronic regions, and 21% in exons or UTRs, consistent with previous results implicating UNC-75 in alternative splicing (*Kuroyanagi et al., 2013a*; *2013b*; *Norris et al., 2014*). We determined overrepresented motifs for UNC-75 binding (*Figure 2B*). The most enriched motif was UGUGUGUG, as exemplified by the binding site on *unc-75* mRNA (*Figure 2C*), consistent with the UNC-75 binding site (G/U)UGUUGUG previously inferred from RNA-seq (*Kuroyanagi et al., 2013b*) and the U(G/A)UUGUG consensus motif defined by RNAcompete (*Norris et al., 2014*). The second most enriched motif G/CAAAAAA is not previously known, and is exemplified by *nrx-1*, a known UNC-75 target (*Kuroyanagi et al., 2013b*; *Norris et al., 2014*) (*Figure 2C*). The list of putative UNC-75 targets identified in our CLIP-seq analysis showed significant overlap with those identified in comparisons of whole-organism transcriptomes of wild type and *unc-75* mutants (*Kuroyanagi et al., 2013b*; *Norris et al., 2014*) (*Supplementary file 2*). Such partial overlap is anticipated given the different techniques used (CLIP-seq of neuronal transcripts vs RNA-seq of the entire organism; see Discussion).

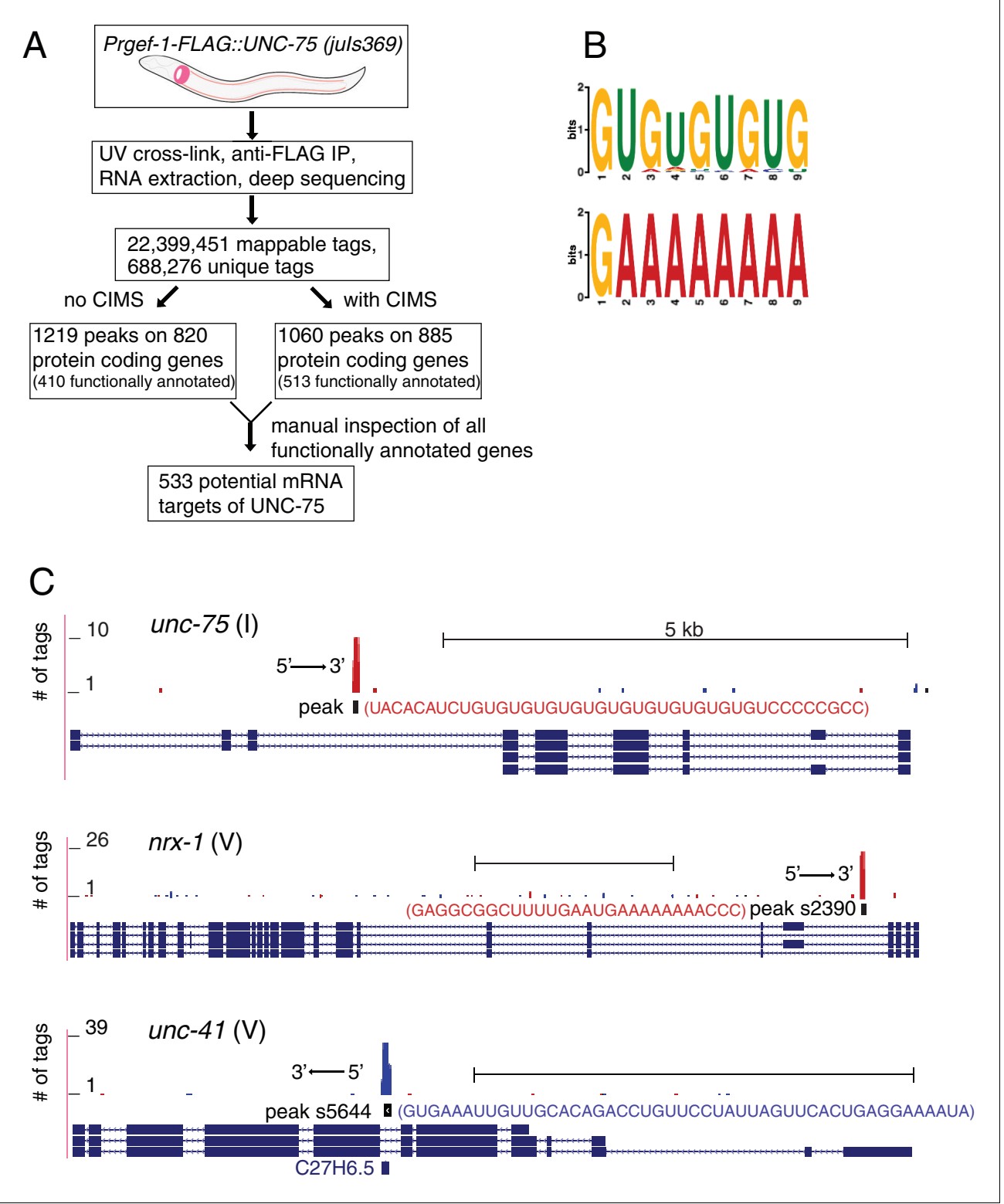

**Figure 2.** CLIP-seq of UNC-75 in *C. elegans* neurons. (**A**) Flow chart of CLIP-seq analysis that identified 533 potential mRNA targets bound by UNC-75. (**B**) The top two motifs enriched in UNC-75 CLIP-seq peaks, based on MEME analysis of the sequence of bound mRNAs. (**C**) UNC-75 CLIP-seq peaks in *unc-75, nrx-1*, and *unc-41*, displayed using the UCSC Genome Browser. Splicing variants are shown under each genomic locus. Sequence of the annotated CIMS peak is shown next to the peak. 's' in peak number stands for 'substitution'. Red and blue tags indicate the two different gene orientation on chromosomes. Scale, 5 kb.

*Figure 2 continued on next page*

*Figure 2 continued*

The following figure supplement is available for figure 2:

**Figure supplement 1.** Purification of UNC-75 and CELF2 bound RNA using CLIP.

When manually checking the genomic loci of UNC-75 targets, we observed that UNC-75 binding sites in introns often overlapped with regions that express snoRNAs. For example, peak s5644 (s stands for substitution of CIMS), located in an intron of *unc-41,* overlaps with the snoRNA C27H6.5 (*Figure 2C*). Although from the CLIP-seq data we cannot determine whether UNC-75 binds both mRNA and snoRNA, it is likely that UNC-75 at least binds to the mRNA, as the mapped reads included nucleotides outside the snoRNA.

We performed Gene Ontology (GO) analysis on the 533 potential protein coding targets of UNC-75 using DAVID (*Huang da et al., 2008*). The three most enriched functional annotation clusters are alternative splicing (49 genes, $P$ = 3.9E-20, Fisher's exact test), nucleotide-binding (60 genes, $P$ = 4.5E-17) and transmembrane (66 genes, $P$ = 2.2E-12), consistent with previously reported functions of UNC-75 in alternative splicing, and validating the quality of our CLIP-seq. The most significantly enriched signaling network is MAPK pathway (9 genes, $P$ = 4.7E-3).

Having identified many mRNA targets bound by UNC-75, we sought targets involved in axon regeneration by taking advantage of our previous genetic screen (*Chen et al., 2011*). Among the 533 protein-coding target genes of UNC-75, we found 17 genes previously identified as required for axon regeneration (*Supplementary file 2*). GO analysis on these 17 genes resulted in the most significantly enriched cluster as cholinergic synaptic transmission ($P$ = 1.6E-6, 4 genes including *unc-32, unc-41, unc-64, unc-75* and *unc-104*). Mutants of these genes have normal PLM axon outgrowth in development and display significantly reduced PLM regrowth (*Chen et al., 2011*), yet their injured PLM axons were able to form regenerative growth cone-like structures more often than wild type axons (*Figure 1—figure supplement 2*). Thus, these synaptic transmission genes, like *unc-75*, appear to be specifically required for efficient regenerative axon extension.

## UNC-75 promotes expression of a neuronal UNC-64/Syntaxin isoform, and represses a non-neuronal UNC-64/Syntaxin isoform

To understand how UNC-75 regulates its targets, we chose *unc-64*/Syntaxin for further study, as a prominent UNC-75 CLIP-seq peak mapped to the last intron (intron 7) of *unc-64* (*Figure 3A*). Alternative splicing of *unc-64* exons 8a and 8b, which flank intron 7b, generates transcripts encoding UNC-64A and UNC-64B isoforms, which differ in their C-terminal hydrophobic membrane anchors (*Ogawa et al., 1998*; *Saifee et al., 1998*). UNC-64A is expressed predominantly in neurons and in some non-neuronal tissues, whereas UNC-64B is only expressed in non-neuronal tissues (*Saifee et al., 1998*) (*Figure 3B,C*). RT-qPCR analyses showed that *unc-64A* mRNAs were significantly reduced in *unc-75* mutants, whereas *unc-64B* mRNAs were increased (*Figure 3—figure supplement 1A*). Total *unc-64* mRNA levels were reduced in *unc-75* mutants compared to wild type, and expression of UNC-64 proteins in neurons was strongly reduced as determined by immunostaining using antibodies that recognize both isoforms (*Figure 3—figure supplement 1B*). Thus, UNC-75 is required for neuronal expression of UNC-64/Syntaxin isoforms.

To dissect how UNC-75 differentially regulates the *unc-64A* and *unc-64B* isoforms we next examined *unc-64* isoform-specific reporters, in which GFP was fused to exon 8a or exon 8b (*Ogawa et al., 1998*; *Saifee et al., 1998*). In *unc-75* mutants expression of an UNC-64A::GFP reporter was strongly reduced (*Figure 3B*), whereas an UNC-64B::GFP reporter was ectopically expressed in the nervous system (*Figure 3C*). To define the roles of UNC-75 binding sites in this differential regulation, we then generated a neuronal splicing reporter for UNC-64A and UNC-64B, using *unc-64* genomic DNA from exon 7 to exon 8b, with RFP inserted 3' to exon 8a and GFP inserted 3' to exon 8b (*Figure 3E*). In a wild type background, these transgenic animals expressed RFP strongly in the nervous system (UNC-64A-like expression), whereas GFP (reflecting the UNC-64B isoform) was undetectable. In *unc-75* mutants, neuronal RFP expression was greatly reduced and GFP expression was increased in neurons (*Figure 3E*). Furthermore, deletion of the UNC-75 CLIP site in intron 7 of the splicing reporter increased UNC-64B::GFP expression in neurons and in non-neuronal tissues (*Figure 3D*). In addition, by RNA-seq, we detected significantly more reads

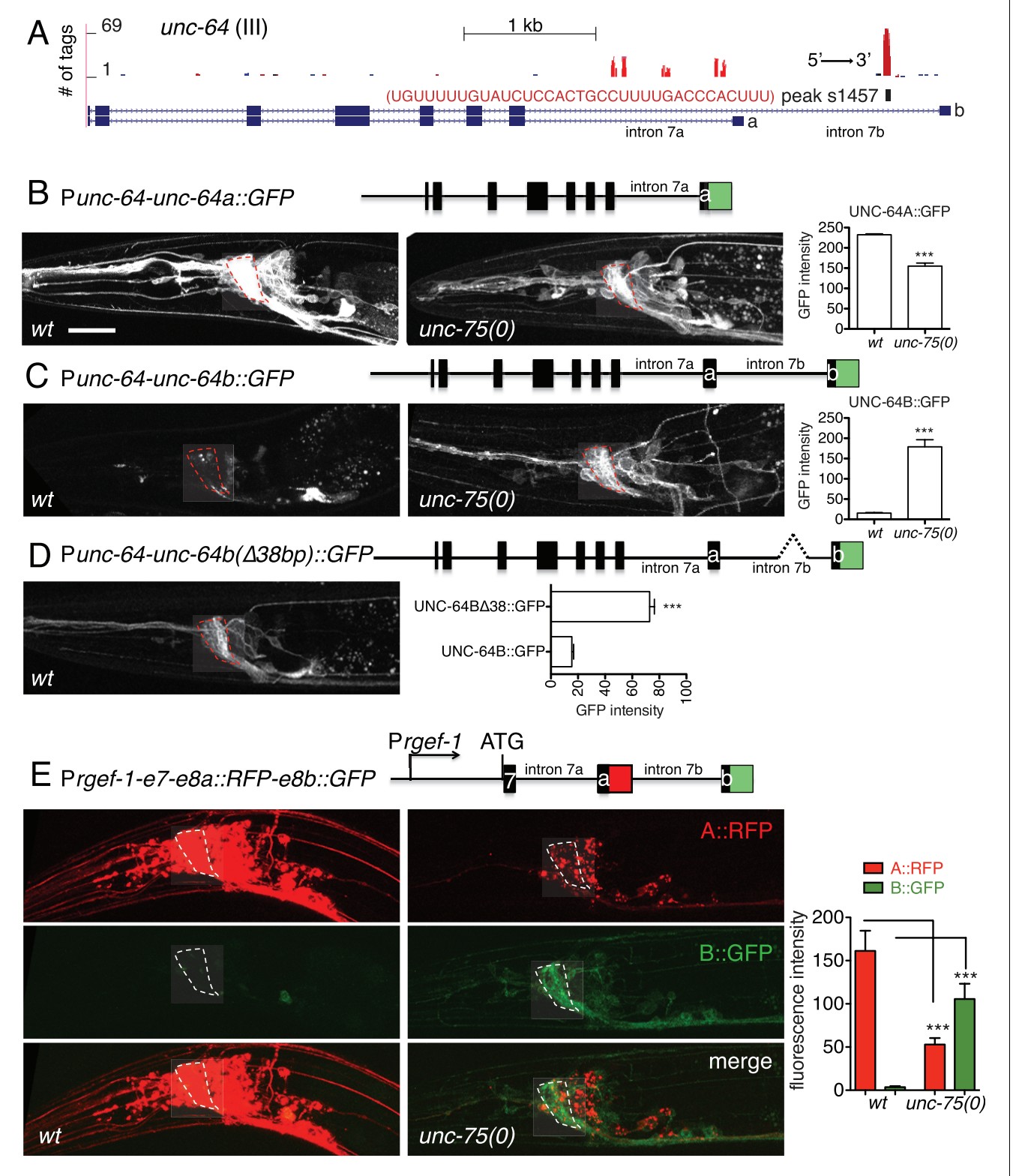

**Figure 3.** UNC-75 regulates alternative splicing of *unc-64*/Syntaxin in neurons. (**A**) UNC-75 CLIP-seq peaks in the *unc-64* locus; genomic track display from UCSC Genome Browser. (**B**) In wild-type animals, UNC-64A::GFP is strongly expressed in most neurons. In *unc-75(0)* mutants UNC-64A::GFP is expressed at lower levels in most neurons. Images of nerve ring and head neurons. (**C**) UNC-64B::GFP is not expressed in the nervous system in wild-type background but is ectopically expressed in head neurons in *unc-75(0)*. (**D**) Deletion of the 38 bp UNC-75 binding site in intron 7 results in neuronal expression of UNC-64B::GFP in wild-type background. (**E**) Images of anterior portion of animals expressing the UNC-64A/B splicing reporter in wild-

*Figure 3 continued on next page*

*Figure 3 continued*

type and *unc-75(0)* mutant backgrounds. *unc-75(0)* is *e950*. The splicing reporter contains the 3' part of *unc-64* genomic sequence (boxes are exon7, 8a and 8b, lines are intron 7a and 7b); RFP is inserted at the end of exon 8a and GFP inserted at the end of exon 8b. For B-E, scale bar 20 μm. Bar charts show quantitation of fluorescence intensity in the nerve ring region (ROIs shown in dashed boxes). Statistics: Student's t-test. N=5–10.

The following figure supplements are available for figure 3:

**Figure supplement 1.** UNC-64 expression is regulated by UNC-75.

**Figure supplement 2.** *unc-64* RNA splicing in wild type and *unc-75* mutants.

**Figure supplement 3.** UNC-64A and UNC-64B have distinct roles in neuronal function.

mapping to intron 7a of *unc-64* in *unc-75* mutants, and fewer reads mapping to intron 7b compared to wild type (*Figure 3—figure supplement 2*). These analyses suggest UNC-75 binding in intron 7 promotes alternative splicing of exon 8a and represses inclusion of exon 8b through intron retention, resulting in neuronal expression of UNC-64A and repression of UNC-64B.

The two isoforms of UNC-64 are both localized to the plasma membrane via their transmembrane domains at the C terminus. A premature stop codon mutation (*js116*) in the UNC-64A transmembrane domain causes complete loss of function, indicating the importance of this domain (*Saifee et al., 1998*). Consistent with this, expression of UNC-64 lacking a transmembrane domain (UNC-64ΔTM) was unable to rescue the movement defect of *unc-64(md130)* (*Figure 3—figure supplement 3*), a partial loss of function (*plf*) allele affecting both isoforms (*van Swinderen et al., 1999*). We further expressed specific isoforms using cDNAs, and found that pan-neuronal expression of UNC-64A, but not of UNC-64B, was able to rescue the locomotion defects of *unc-64(plf)* (*Figure 3—figure supplement 3*). Thus UNC-64A appears to be the major functional Syntaxin isoform in neurons; even when ectopically expressed in neurons, as in *unc-75* mutants, UNC-64B is unable to substitute for UNC-64A.

## UNC-64 is specifically required for axon regeneration but not for axon development

Partial loss of function in *unc-64 (md130* or *e246)* results in a partial block in PLM axon regeneration (*Figure 4A*). As *unc-64(js115)* null mutants arrest in the first larval stage (*Saifee et al., 1998*), we used two approaches to examine the null phenotype of *unc-64* in regeneration. We first tested *unc-64(js115)* null mutants in which lethality, but not movement, was rescued by expression of UNC-64 under the combined control of *acr-2*, *unc-17*, and *glr-1* promoters (*Hammarlund et al., 2007*). In such animals PLM developed normally and axon regeneration was inhibited to a similar extent as in *unc-64(plf)* (*Figure 4A*). To address caveats due to possible misexpression of this transgene in touch neurons, we also generated a single-copy transgene containing LoxP-flanked *unc-64A(cDNA)* driven by the pan-neural *rgef-1* promoter, which fully rescued *unc-64(0)* lethality and locomotor defects (*Figure 4B,C*). We then excised the floxed copy of *unc-64A(+)* in touch neurons using P*mec-7*-nCre (*Chen et al., 2015*). The *unc-64(0)* PLM axons developed normally (*Figure 4—figure supplement 1*) and displayed reduced regrowth after axotomy, comparable to *unc-64(plf)* (*Figure 4A*). We conclude that UNC-64 is specifically required for axon regeneration but not development, and that the null phenotype of *unc-64* is a partial block in regeneration. Regenerative growth cones in *unc-64(md130)* mutants had dynamic filopodia but did not effectively elongate, resembling those of *unc-75(0)* (*Video 3*). Moreover, regrowth defects in *unc-64(md130)* were rescued by pan-neuronal expression of either UNC-64A or UNC-64B (*Figure 4A*), suggesting either isoform is sufficient for function in regeneration when overexpressed.

## UNC-64A overexpression suppresses *unc-75* axon regeneration and locomotion defects

To test the hypothesis that altered UNC-64 expression underlies the regeneration defects in *unc-75* mutants, we overexpressed UNC-64 in *unc-75* mutants and examined axon regrowth. Strikingly, transgenes containing *unc-64* genomic DNA, which produce both UNC-64A and UNC-64B,

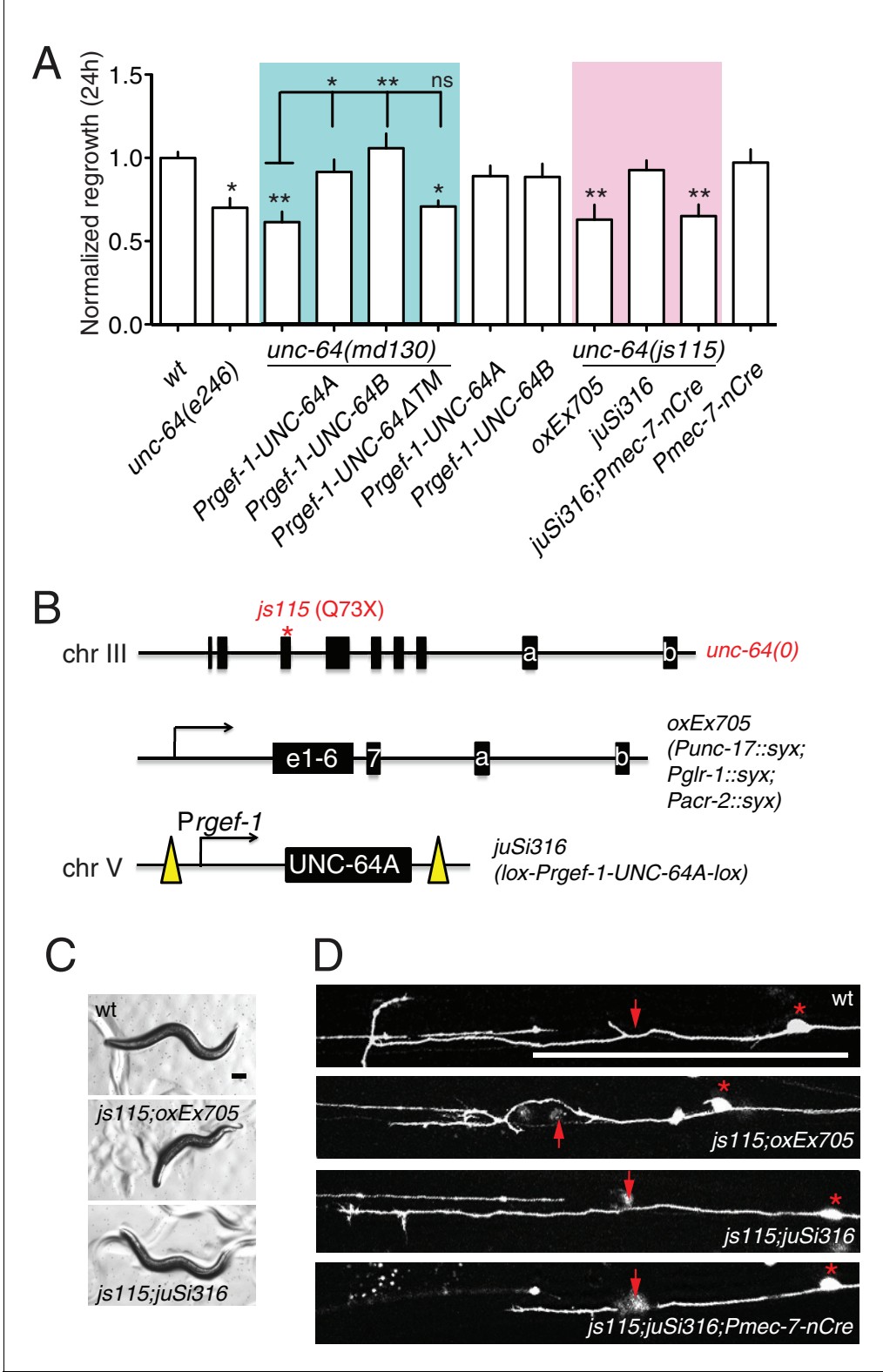

**Figure 4.** *unc-64* is required cell autonomously for PLM axon regeneration. (**A**) Normalized PLM regrowth 24 hr post axotomy. PLM axon regeneration is reduced in mutants with *unc-64* partial loss of function alleles *md130* and *e246*, as well as null allele *js115*. These alleles affect both isoforms (*Saifee et al., 1998*). *unc-64(md130)* regeneration phenotypes are rescued by pan-neural expression of UNC-64A or UNC-64B, but not by UNC-64ΔTM. Expression of UNC-64A or B in a wild type background does not affect PLM regeneration. Statistics, One-way ANOVA followed by Bonferroni's Multiple Comparison Post Test. N ≥ 10. (**B**) Schematic illustration of two strategies to generate *unc-64* mutation in touch neurons. The
*Figure 4 continued on next page*

*Figure 4 continued*

lethality of *unc-64(js115)* is rescued by *oxEx705* or *juSi316*. *juSi316* was crossed to *Pmec-7-nCre* to delete transgenic UNC-64 in touch neurons. (**C**) Representative images of animals with indicated genotypes. *js115; oxEx705* animals were viable but severely Unc, while *js115; juSi316* animals were viable and slightly Unc. (**D**) Representative PLM regrowth images 24 hr post axotomy. Asterisks: PLM cell body; red arrow, injury site. Scale bar: 100 μm.

The following figure supplement is available for figure 4:

**Figure supplement 1.** UNC-64 is not required for PLM development.

significantly suppressed the PLM regeneration defects of *unc-75* mutants (***Figure 5A,B***). Furthermore, pan-neuronal overexpression of UNC-64A (using cDNA), but not of UNC-64B, rescued *unc-75* regeneration defects to a similar degree as *unc-64* genomic DNA, whereas UNC-64ΔTM did not rescue (***Figure 5A,B***). Moreover, transgenes expressing *unc-64* genomic DNA or UNC-64A strongly suppressed *unc-75* locomotor phenotypes, while overexpression of UNC-64B or UNC-64ΔTM did not rescue (***Figure 5C,D***). We infer that decreased neuronal expression of the UNC-64A isoform is a major contributor to *unc-75* mutant phenotypes in regeneration and behavior.

## CELF RBPs play a conserved role in axon regeneration

The CELF protein family is highly conserved from *C. elegans* to humans. By sequence comparison, UNC-75 is more closely related to the CELF3/4/5/6 subfamily (***Figure 6—figure supplement 1A***), and a previous study showed that expression of human CELF4 partly rescues the locomotion defects of *unc-75* mutants (***Loria et al., 2003***). Here, we found that either mouse CELF2 or CELF4, when expressed in *C. elegans* touch neurons, could significantly rescue the *unc-75* PLM regrowth defect, with CELF2 being slightly more effective than CELF4 (***Figure 6A***). This may be because CELF2, like UNC-75, is generally localized in the nucleus in neurons (***Otsuka et al., 2009***) whereas CELF4 is predominantly cytoplasmic (***Wagnon et al., 2012***). Nonetheless, this result suggests that the CELF proteins may also play a conserved role in axon regeneration.

Among mouse *Celf* genes, *Celf2* and *Celf4* are most highly expressed in the postnatal CNS (***Supplementary file 3***). We further examined *Celf2* and *Celf4* expression in dorsal root ganglion (DRG) neurons using RT-qPCR. *Celf4* transcripts increased during postnatal development (***Figure 6B***), consistent with reported expression (***Supplementary file 3***). In contrast, *Celf2* mRNA levels declined from perinatal stages to adult (***Figure 6B***). Notably, *Celf2* transcript levels significantly increased in DRGs after sciatic nerve crush in 2-month old animals (***Figure 6C***). As the peripheral processes of DRG neurons are capable of regeneration, and such capacity declines with age (***Mar et al., 2014***), the developmental decline in *Celf2* expression and its upregulation after injury suggest *Celf2* expression correlates with axon regeneration capacity.

## CELF2 is required for sciatic nerve regeneration

To test the function of CELF2 in mouse axon regeneration, we generated a conditional allele of *Celf2* in which exon 3 was flanked by loxP sites (***Figure 6—figure supplement 1B***). Exon 3 encodes part of RRM1, a domain present in most CELF2 isoforms and required for protein function (***Singh et al., 2004***). Cre-mediated deletion of exon 3 would alter splicing, resulting in *Celf2* mRNA encoding non-functional CELF2 proteins due to frameshift followed by premature stop (***Figure 6—figure supplement 1B***). We expressed Cre recombinase ubiquitously in the *Celf2^{flox}* background (see Materials and methods) to generate a *Celf2^-* allele, and verified the genomic deletion of exon 3. These constitutive *Celf2^{-/-}* mice died neonatally, with 5% escapers surviving up to 3 weeks. No obvious morphological defects were detected in newborn *Celf2^{-/-}* animals by histological analysis (not shown). To test the role of *Celf2* in axon growth, we assayed cultured primary neurons. Explants of DRG neurons derived from E13.5 *Celf2^{-/-}* embryos displayed significantly reduced axon extension (***Figure 6—figure***

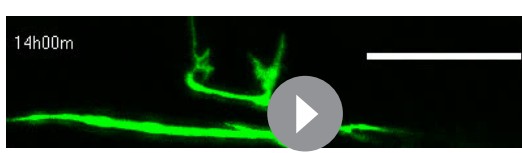

**Video 3.** Time-lapse movie of the tip of a regenerating PLM axon from an *unc-64(md130)* mutant, starting at 14 hr post axotomy, ending at 15 hr post axotomy.

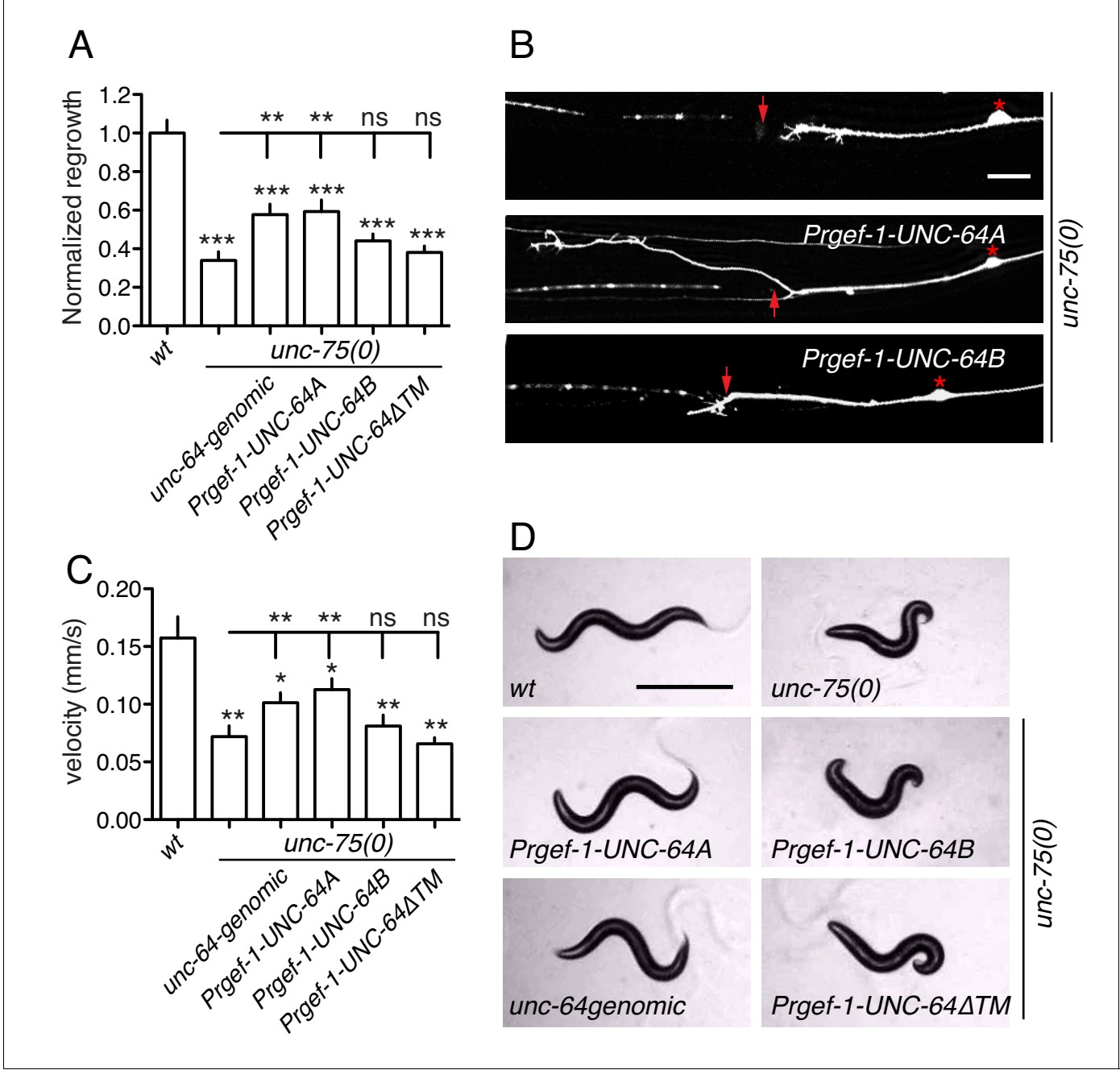

**Figure 5.** Overexpression of UNC-64A suppresses *unc-75* neuronal phenotypes. (**A-B**) Quantitation (n ≥ 10 per genotype) and images of PLM axon regeneration 24 hr post axotomy; scale, 10 μm. Asterisks: PLM cell body; red arrow, injury site. Transgenic UNC-64 expression using 9 kb genomic DNA encoding both UNC-64A and UNC-64B was able to partially rescue the regeneration defect of *unc-75(0)*. Pan-neuronal expression of UNC-64A cDNA, but not UNC-64B, significantly increased axon regrowth in *unc-75(0)* mutants. (**C-D**) Quantitation of locomotion velocity and images of animals with indicated genotypes; scale, 0.5 mm. N ≥ 10 per genotype; statistics: One way ANOVA with Bonferroni post test. Genomic DNA or UNC-64A cDNA driven by Pan-neuronal promoter partially rescues the *unc-75* locomotor phenotype; expression of UNC-64B or UNC-64ΔTM cDNA does not rescue. *e950* is used in *unc-75(0)*.

*supplement 1C,D*). Thus although *Celf2* does not seem to play a major role in developmental axon outgrowth, it is required for efficient regrowth of axons from explants.

　　To study CELF2 function in adult axon regeneration, we next generated a nervous system conditional knockout by crossing the *Celf2(flox)* allele to a *Nestin-Cre* line (***Tronche et al., 1999***). *Nestin-*

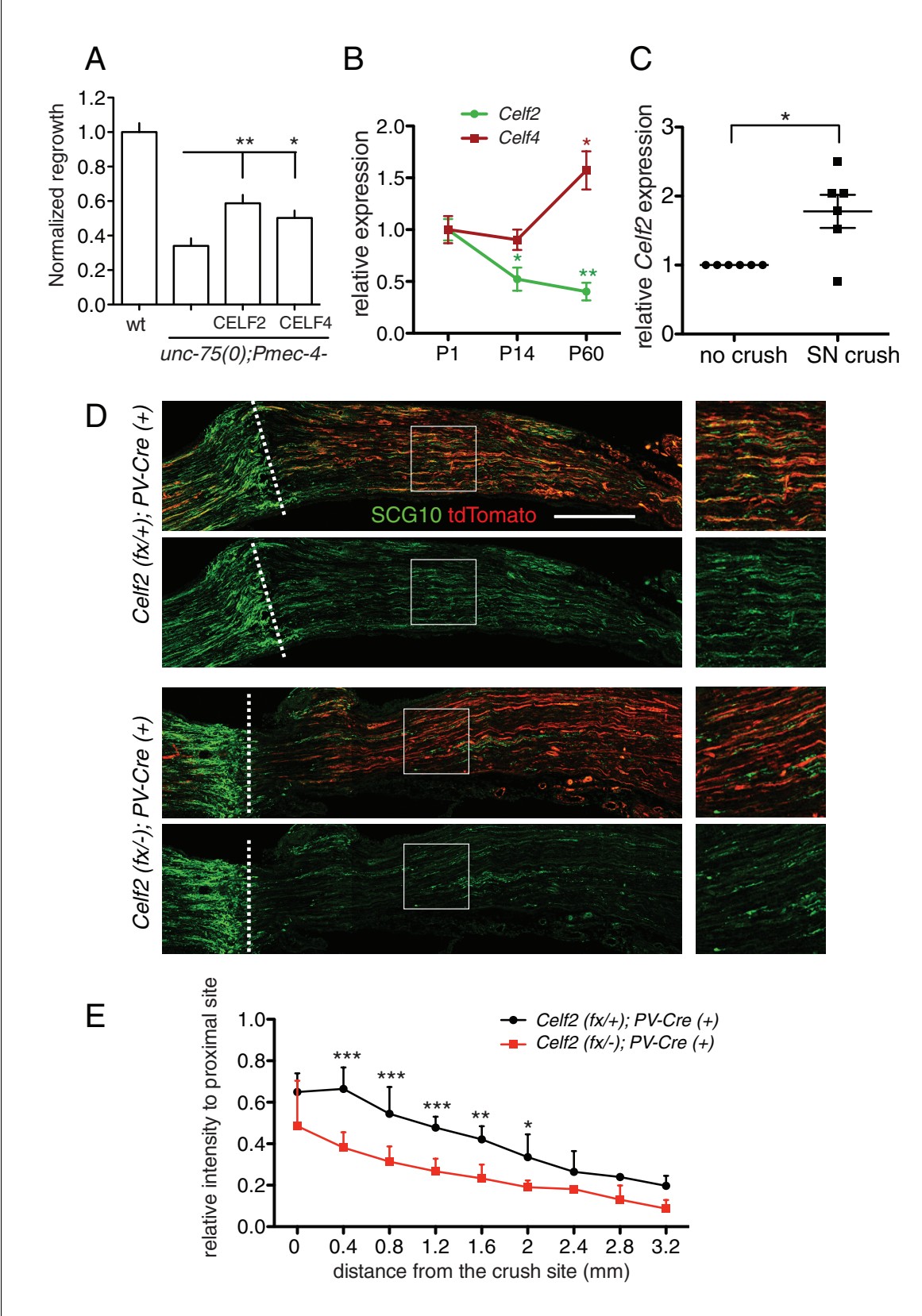

**Figure 6.** CELF proteins play conserved roles in axon regeneration. (**A**) Expression of full length cDNAs of mouse CELF2 or CELF4 with mCherry tag in *C. elegans* touch neurons significantly rescues the *unc-75* axon regrowth defect. Normalized PLM axon regrowth 24 hr post axotomy, N = 14–28. Based

*Figure 6 continued on next page*

*Figure 6 continued*

on mCherry fluorescence the CELF2 and CELF4 transgenes are expressed at similar levels (not shown). (**B**) *Celf2* transcript levels in DRG neurons decrease during postnatal development, whereas *Celf4* levels increase. Expression was normalized to P1, and mouse β-*Actin* was used as internal reference. Statistics, One-way ANOVA followed by Bonferroni's Multiple Comparison Post Test. N = 4–6. (**C**) Expression of *Celf2* transcripts in DRG of 2 month old mice is significantly enhanced 3 days after sciatic nerve injury. Ratio of the crushed side to the uncrushed side within the same animal is plotted. Statistics, Student's t-Test. (**D**) Mutation of *Celf2* impairs axon regeneration in DRG PV+ neurons. Confocal images of longitudinal sciatic nerve sections distal to the lesion, stained with anti-SCG10 (green) at 3 days post crush. tdTomato expression is from Rosa26-lox-STOP-lox-tdTomato and marks neurons in which Cre was active. Enlarged images of the boxed areas are shown on the right; white dashed line marks the lesion site. Scale bar: 0.5 mm. (**E**) Quantitation of SCG10 intensity in tdTomato positive axons at different distances from the lesion site, normalized to SCG10 intensity proximal to the lesion. 6 control and 5 mutant animals were analyzed. Statistics, One-way ANOVA followed by Bonferroni's multiple comparison post test.

The following figure supplement is available for figure 6:

**Figure supplement 1.** *Celf2* is required for neurite growth in mouse DRG neurons.

*Cre* induced *Celf2* knockout animals were smaller than littermates (***Figure 6—figure supplement 1E***) and usually survived 1–2 months. Using an in vitro regeneration assay (***Saijilafu and Zhou, 2012***), we found *Celf2* knockout neurons showed defects in adult DRG regeneration after re-suspension and re-plating of DRG cells (***Figure 6—figure supplement 1F,G***). To examine in vivo regeneration, we crossed *Celf2^flox^* allele to a *Parvalbumin-Cre* driver to delete *Celf2* in parvalbumin-expressing large diameter DRG neurons, which make up ~30% of DRG axons (***Hippenmeyer et al., 2005***). *Celf2^flox/-^; parvalbumin-cre^+/-^* mice were superficially wild-type, allowing us to examine axon regeneration in adult stages. We used *R26/CAG^tdTomato^* (***Madisen et al., 2010***) to label cells with Cre induced *Celf2* deletion. We performed sciatic nerve crush on 2 months old animals and evaluated DRG axon regeneration 3 days post injury by staining for the regeneration marker SCG10 (***Shin et al., 2014***). SCG10 positive DRG axons distal to the crush site were significantly reduced in the *Celf2* mutant compared to littermate controls (***Figure 6C,D***), indicating that CELF2 is required for DRG axon regeneration. Together, these data support a conclusion that CELF proteins play a conserved role in axon regeneration from *C. elegans* to mouse.

## CELF2 can regulate expression of multiple Syntaxins

CELF2 is known as a splicing regulator, but its role in the nervous system has not been explored. To test whether CELF2 regulates similar sets of target mRNAs as UNC-75, we performed CLIP-seq of mouse CELF2 using a neuroblastoma N2A cell line (http://www.atcc.org/products/all/CCL-131.aspx#generalinformation) that stably expresses BLRP (biotin ligase recognition peptide) tagged CELF2 (***Figure 2—figure supplement 1B***). We identified 2919 protein coding genes as CELF2 targets (***Supplementary file 4***) and 'UGUGUGUG' as the most significant binding motif, which is conserved to UNC-75, suggesting the function of CELF genes in target regulation is highly conserved.

As *C. elegans* UNC-75/CELF regulates UNC-64/Syntaxin alternative splicing, we asked whether CELF-Syntaxin regulation might be conserved in mammals. From CELF2 CLIP-seq we identified CELF2 binding sites in multiple mouse *Syntaxin* genes (***Figure 7A*** and ***Supplementary file 4***). We examined expression of candidate *Syntaxin* genes in *Celf2* knockout mouse brain by RT-qPCR, and observed that mRNA levels of splicing variants of *Syntaxin2* and *Syntaxin16* were significantly altered in *Celf2^-/-^* mouse brain (***Figure 7B***). Syntaxin2 is ubiquitously expressed (***Bennett et al., 1993***), but its role in the nervous system is not known. Reminiscent of *unc-64*, *Syntaxin2* is alternatively spliced, generating two isoforms differing in the C-terminal membrane anchor. From CLIP-seq we found a CELF2 binding site in the intron 5' to the alternatively spliced exons (***Figure 7A***). Syntaxin16 expression is enriched in neurons in the brain and has been implicated in neurite growth (***Chua and Tang, 2008***). Two splicing variants of *Syntaxin16* differ in exons encoding the N terminus, which is known to interact with Vps45 (***Dulubova et al., 2003***). We detected CELF2 binding on the introns in the immediate vicinity to the alternatively spliced exons, as well as on the 3' UTR (***Figure 7A***). Taken together, these data support a hypothesis that CELF-mediated regulation of alternative splicing of Syntaxin genes is likely a conserved mechanism in axon regeneration.

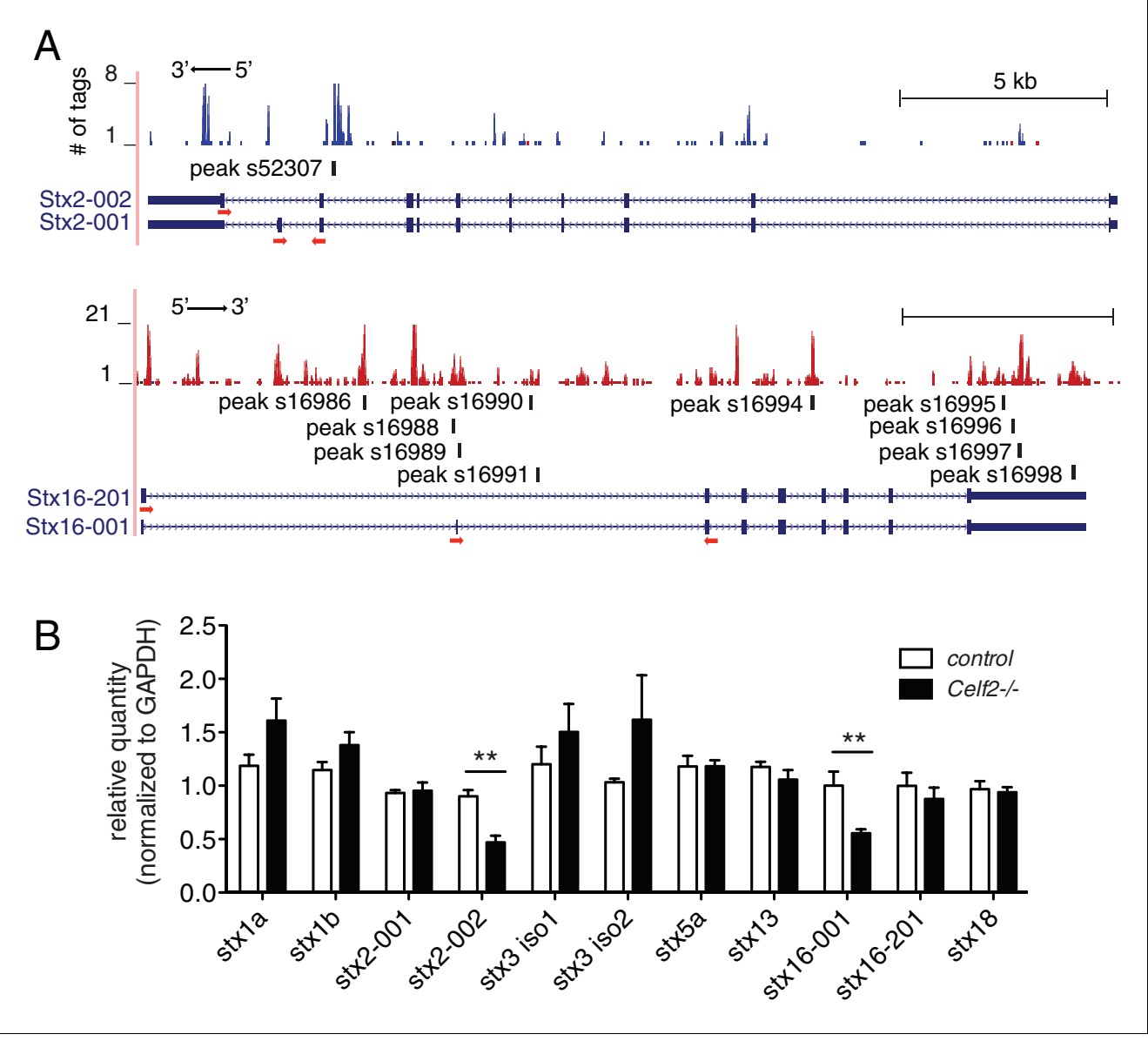

**Figure 7.** CELF2 regulates expression of specific neuronal Syntaxin isoforms. (**A**) Genome browser tracks displaying CELF2 CLIP-seq peaks on the *stx2* and *stx16* gene loci. Red and blue tags indicate the two different gene orientation on chromosomes. CELF2 binding peaks are mapped to introns near alternatively spliced exons. Isoform labeling is consistent to gene annotation on Ensembl. Red arrows under the exons implicate primers used for RT-qPCR in panel b. (**B**) Transcript levels of Syntaxin genes were measured by RT-qPCR in E15.5 control and *Celf2⁻/⁻* brains. Statistics, Student's t-test, N = 5–6. Expression of the alternatively spliced isoform 002 of *stx2* and isoform 001 of *stx16* was significantly decreased in *Celf2⁻/⁻* constitutive mutants.

## Discussion

We have shown that CELF RNA-binding proteins play conserved roles in axon regeneration from *C. elegans* to mammals. Using genomic approaches we identify synaptic transmission genes as key CELF targets. Expression of neuronal Syntaxins is regulated by CELF-mediated alternative splicing and is critical for regeneration. The CELF/Syntaxin pathway is not required for regenerative growth cones to form but allows them to efficiently extend axons. We propose that post-transcriptional regulation by CELF balances the production of mRNA and protein isoforms during axonal regeneration.

## UNC-75 targets in *C. elegans* neurons

CELF proteins are long known as regulators of RNA splicing and stability, but their roles in neuronal RNA regulation are only recently being explored. Our findings, together with previous RNA-seq analyses, identify contexts in which UNC-75/CELF regulates alternative splicing for inclusion of neuron-specific exons in *C. elegans*. The production of neuronal mRNA encoding *unc-32*/ATPase depends on alternative splicing of two sets of mutually exclusive exons, which involves UNC-75 binding to their flanking introns, partly in collaboration with the RBPs ASD-1 and FOX-1 (*Kuroyanagi et al., 2013a*). Motor neuron subtype alternative splicing of *unc-16*/JIP3 mRNA also involves UNC-75 binding to intron sequences, and the outcome of exon inclusion or exclusion partly depends on the Elav-like RBP EXC-7 in cholinergic neurons (*Norris et al., 2014*). These previous studies were based on comparisons of whole-organism RNA levels, which reflect both direct and indirect regulation by UNC-75. As neurons contribute a small fraction of the overall *C. elegans* transcriptome, many neuronal targets of UNC-75 may have been overlooked in such studies. Our use of neuronal CLIP-seq validates some previous targets as direct interactors of UNC-75 and has expanded the number of candidate targets by an order of magnitude.

Many of UNC-75's neuronal targets are involved in synaptic transmission, and were also identified as required for axon regeneration (*Chen et al., 2011*). The close similarity between *unc-75* and *unc-64* regeneration phenotypes, and the ability of UNC-64 overexpression to rescue *unc-75* defects suggest that they act in a common pathway for regenerative axon extension. We therefore focused on how UNC-75 mechanistically promotes UNC-64 expression. Our analysis shows that UNC-75 binding to intron 7b promotes inclusion of the upstream exon 8a, leading to production of UNC-64A at the expense of UNC-64B. These results support the general model that UNC-75 tends to promote inclusion of upstream exons by binding downstream intronic sites (*Kuroyanagi et al., 2013b*; *Norris et al., 2014*). Loss of UNC-75 caused increased pre-mRNA containing *unc-64* intron 7a and decreased pre-mRNA containing intron 7b, suggesting that alternative splicing of *unc-64* is regulated via selective intron retention. In *unc-75* mutants, we detected reduced UNC-64A::GFP expression from our reporter line (P*unc-64-unc-64A*::gfp), although the reporter lacked intron 7b (*Figure 3B*). The reduction of UNC-64A expression from this reporter was not as dramatic as that from the dual reporter containing intron 7b (*Figure 3E*). We speculate that if the primary binding site is not available (i.e. intron 7b), UNC-75 can bind to secondary sites (e.g. intron 7a) to regulate UNC-64A expression. Consistent with this idea, we detected UNC-75 binding tags from our CLIP-seq in intron 7a (*Figure 3A*).

## Functions and targets of CELF2 in neurons

CELF2 was identified almost 20 years ago, in multiple studies, first as a CUG repeat binding protein CUG-BP2 (*Timchenko et al., 1996*); in screens for homologs of the Elav family (ETR-3) (*Lu et al., 1999*); and as a gene induced in apoptotic neuroblastoma cells (NAPOR) (*Choi et al., 1998*). However until now, in vivo functions of CELF2 have not been explored using genetics. We find that CELF2 is essential for viability. The cellular basis of the lethal phenotype remains to be determined, but may reflect CELF2 functions in non-neuronal cells, as animals with a neuronal deletion of CELF2 were viable, albeit smaller and short-lived compared to wild type.

Alternative splicing (AS) is pervasive in neurons and has been implicated in axon guidance, synaptogenesis and synaptic transmission (*Raj and Blencowe, 2015*). The function of CELF family members in AS was first explored in non-neuronal cells (e.g. cardiomyocytes), but given their widespread expression in the brain increasing attention has been given to the neuronal roles of CELF proteins (*Ladd, 2013*). CELF4 appears to be predominantly expressed in the cytoplasm of neurons and is implicated in neuronal excitability (*Wagnon et al., 2012*; *Yang et al., 2007*). A CELF4 CLIP-seq analysis in mouse brain identified numerous targets, including many genes implicated in synaptic transmission, although CELF4 is thought to primarily regulate mRNA translation (*Wagnon et al., 2012*). CELF2 is known to regulate the AS of specific neuronal genes, including NMDAR1, MAPT/tau, and NF1 (*Barron et al., 2010*; *Han and Cooper, 2005*; *Leroy et al., 2006*). Screens for CELF2 targets have been performed using SELEX (*Faustino and Cooper, 2005*) and by CELF2 overexpression in T-cells (*Mallory et al., 2015*). Here we report a genome-wide CLIP-seq approach to find CELF2 targets in neuronal cells. Consistent with our analysis of UNC-75 neuronal targets, in neuroblastoma cells, CELF2 binding sites were found in several Syntaxin genes. Like UNC-64, several mammalian

Syntaxins are known to undergo alternative splicing, and we find that expression of two Syntaxin isoforms (stx2-002 and stx16-001) is significantly reduced in *Celf2* mutant brains.

## Regulation of CELF expression and function

We find that CELF2 expression is upregulated after nerve injury. Expression of CELF2 or other CELF genes is altered by cellular stress or damage in several contexts, although the mechanisms of induction remain poorly understood. In epithelial cells CELF2 expression is induced by UV or gamma irradiation injury (*Mukhopadhyay et al., 2003*). In the nervous system CELF1 has been reported to be upregulated after spinal cord injury (*Yang et al., 2015*). CELF2 expression is also downregulated in some models of ischemic brain injury and fetal alcohol syndrome, although it is unclear whether this is cause or consequence (*Otsuka et al., 2009*). Acute induction of CELF2 expression has been most extensively studied during T cell activation, and occurs partly at the level of transcription and partly by stabilization of *Celf2* transcripts (*Mallory et al., 2011*; *2015*). Recently CELF2 has been found to regulate alternative splicing of the MAPKK MKK7, facilitating a JNK-dependent positive feedback loop during T cell activation (*Martinez et al., 2015*). JNK is required for the increased stability of CELF2 messages after T cell activation, by as-yet unknown mechanisms. Many targets of JNK-regulated AS are also dependent on CELF2 (*Martinez et al., 2015*), suggesting the CELF2/JNK positive feedback loop might function widely to regulate inducible alternative splicing.

## Syntaxin isoforms and synaptic transmission genes function in axon extension

UNC-75 dependent splicing is critical for UNC-64A expression in neurons, at the expense of UNC-64B. The roles of the two UNC-64 isoforms and the mechanistic basis for their functional difference have remained a topic of debate. As truncation of UNC-64A causes apparent complete loss of function (*Saifee et al., 1998*), UNC-64A appears to be the predominant functional isoform in neurons. Overexpression of either A or B isoform rescued the axon regrowth defect of *unc-64* hypomorphs, but only the A isoform was sufficient to rescue *unc-64* locomotor defects when overexpressed. The level of UNC-64 function required in regeneration may be lower than in behavior. A recent study also reported *unc-64* splicing is regulated by *unc-75* (*Norris et al., 2014*). However, using fosmid transgenes, this study found that while either A or B isoform could rescue *unc-64* Ric phenotypes, only UNC-64B could rescue *unc-64* locomotor phenotypes. The basis for this discrepancy might reflect differences in the promoters and nature of expression (pan-neural and cDNA in our study, endogenous regulatory sequences including splicing of 5' exons in Norris et al.). We find that transgenic expression of UNC-64A (cDNA bypassing splicing) also significantly suppressed *unc-75* locomotor and regeneration phenotypes whereas overexpression of UNC-64B was unable to rescue. The simplest interpretation of these data is that UNC-64B is unable to fully substitute for reduced levels of UNC-64A. Possibly, UNC-64B but not UNC-64A requires some additional cofactor lacking in *unc-75*.

The C-terminal membrane anchors of UNC-64A and B are both 25 aa residues long and differ only in the last 14 aa, suggesting that the precise composition of the anchor is critical for Syntaxin function. Both have the same number of hydrophobic residues in their C-termini, with the only discernible difference being that these residues are more clustered in the B isoform. Nevertheless seemingly small differences in hydrophobicity can cause Syntaxins to sort into distinct membrane domains (*Milovanovic et al., 2015*; *Milovanovic and Jahn, 2015*), and may underlie the nonequivalence of the UNC-64A and B isoforms.

UNC-64 is known to localize along the axonal plasma membrane, and may provide a SNARE function involved in membrane addition during axon extension. Like *unc-64, Syntaxin2* encodes two alternatively spliced isoforms that differ in the membrane anchor. Syntaxin2 (also called epimorphin) was first identified as an extracellular morphogen (*Hirai et al., 1992*), but later was also found to function in the cytoplasm as t-SNARE regulating vesicle fusion (*Bennett et al., 1993*). The localization of this protein on either the cytoplasmic face or the extracellular surface of the plasma membrane is determined by its distinct conformation (*Chen et al., 2009*). Little is known of the neuronal functions of Syntaxin2. In mammals Syntaxin16 has been implicated in neurite outgrowth, and appears to localize to the Golgi (*Chua and Tang, 2008*). Mammalian Syntaxin12 (previously called Syntaxin13) has been shown to promote axon regeneration under the control of the mTOR pathway

(*Cho et al., 2014*). Whether other Syntaxins also play a role in axon regeneration remains to be tested.

The mechanism by which Syntaxin contributes to regenerative axonal elongation remains to be elucidated. Because *unc-64* mutants are defective in axon extension rather than growth cone reformation, we believe it is less likely that Syntaxin is required for membrane resealing immediately after injury. Instead, Syntaxin may contribute to the rapid plasma membrane expansion required during axon regrowth (*Bloom and Morgan, 2011*). In mammalian neurons Syntaxin-3 has been implicated in fatty acid stimulated axon plasma membrane expansion resulting from fusion of transport organelles (*Darios and Davletov, 2006*). Such organelles might be related to the Syntaxin-containing transport packets involved in presynaptic assembly (*Ahmari et al., 2000*). The recycling endosome component Syntaxin13 is upregulated by injury and is thought to promote membrane recycling required for regrowth (*Cho et al., 2014*). Finally, Syntaxin might contribute to membrane expansion via a non-fusogenic mechanism as shown for the SNARE Sec22b (*Petkovic et al., 2014*).

In summary, we have revealed a novel regulatory pathway important for axon extension in regenerative regrowth, involving CELF-dependent alternative splicing of neuronal Syntaxins. As well as being relevant to axon regeneration, our finding may shed light on the roles of CELFs in neurological disease. CELF2 is misregulated in a mouse model of spinal muscular atrophy (*Anderson et al., 2004*) and CELF2 copy-number variation has been linked to schizophrenia (*Xu et al., 2011*). Our results add to the conceptual framework for dissecting these complex diseases.

## Materials and methods

### *C. elegans* genetics, molecular biology and transgenes

*C. elegans* were cultured at 15–25°C using standard procedures. Transgenes were introduced into mutants by crossing or injection; homozygosity for all mutations was confirmed by PCR or sequencing. We followed standard procedures to generate new clones and transgenes (All new strains and transgenes are listed in *Supplementary file 5*).

### Fluorescence microscopy and laser axotomy

Fluorescence images were collected using Zeiss LSM710 or LSM510 confocal microscopes. Laser axotomy was performed as previously described (*Chen et al., 2011*). Raw (non normalized) axon regrowth measurements are show in *Supplementary file 7*.

Timelapse movies of PLM axon regrowth were taken with Zeiss LSM710 using agarose beads to immobilize worms. Immunostaining was performed as described (*Saifee et al., 1998*). Briefly, worms were fixed in Bouin's fixative and washed in fresh BTB buffer (1 x Borate Buffer, 0.5% Triton X-100, 2% BME), then stained with UNC-64 antiserum at 1:50 dilution.

### Locomotion analysis

We measured locomotion velocity using WormTracker 2.0 as previously described (*Chen et al., 2015*). Briefly, individual young adults were transferred to a fresh tracking plate with thin OP50 bacteria lawn. 1 min later, the plate was placed on the tracker platform and locomotion recorded for 1 min at 10 frames per second for each animal.

### Crosslinking Immunoprecipitation and deep sequencing (CLIP-seq) in *C. elegans*

CLIP-seq was performed as previously described (*Zisoulis et al., 2010*). CZ14662 [*unc-75(e950)*; P*rgef-1*-UNC-75S(*juIs369*)] worms were crosslinked in a 150 mm plate (with no food) with an energy output of 6 kJ/m$^2$. Worms were lysed by sonication in Homogenization Buffer (100 mM NaCl, 25 mM HEPES, 250 μM EDTA, 2 mM DTT, 0.1% NP-40, 25 units/ml RNasin and Protease Inhibitors). Lysates were centrifuged at 16,000g for 15 min at 4°C and supernatants collected and incubated with M2 magnetic beads (Sigma) overnight on a rotator.

Beads were collected and washed twice with Wash Buffer (1X PBS with 0.1% SDS, 0.5% sodium deoxycholate, and 0.5% NP-40), twice with High Salt Wash Buffer (5X PBS, 0.1% SDS, 0.5% sodium deoxycholate, and 0.5% NP-40) and twice with Polynucleotide Kinase Buffer (PNK Buffer) (50 mM Tris-Cl pH 7.4, 10 mM MgCl$_2$ and 0.5% NP-40). Beads were incubated with 500 μl of Micrococcal

Nuclease Reaction Buffer (50 mM Tris-Cl pH 7.9, 5 mM $CaCl_2$) containing 1 ng of Micrococcal Nuclease (NEB) for a total of 10 min at 4°C with intermittent shaking on a Thermomixer (Eppendorf) (1200 rpm for 1 min and then 1200 rpm for 15 sec every 3 min). Beads were then washed twice with PNK+EGTA Buffer (50 mM Tris-Cl pH 7.4, 20 mM EGTA and 0.5% NP-40), twice with Wash Buffer and twice with PNK Buffer. The beads were incubated for 10 min at 37°C in the Thermomixer with intermittent shaking (1200 rpm for 15 sec every 3 min) in 80 µl NEB Buffer 3 containing 30 units of Calf Intestinal Phosphatase (NEB). The beads were then washed twice with PNK+EGTA Buffer, twice with PNK Buffer and twice with 0.1 mg/ml BSA. An RNA linker (5'- CUCGUAUGCCGUCUUCUGC UUG-3' 3' Puromycin, 5'P) with a puromycin modification at the 3′ end to avoid self-circularization was linked to the mRNA present in the complexes by T4 RNA Ligase and incubated overnight at 16°C with gentle shaking (1300 rpm every 5 min for 15 s in the Thermomixer). The beads were washed three times with PNK Buffer and incubated in 80 µl PNK Buffer (NEB) with 40 units of T4 PNK enzyme (NEB) in the presence of $^{32}$P-γ-ATP (1 mCi). Samples were incubated for 10 min at 37°C with intermittent shaking (1000 rpm every 4 min for 15 s). Cold ATP was added to the reaction at a final concentration of 1.25 mM and incubated for 5 min. The reaction was terminated with three washes of PNK+EGTA Buffer. Complexes were eluted from the beads by incubation for 10 min at 70°C in Nupage LDS Buffer. Samples were loaded onto a native 10% Bis-Tris gel with MOPS SDS Running Buffer. Next, the samples were transferred to a nitrocellulose membrane. The band corresponding to UNC-75/RNA complexes (65–100 kDa) was cut out of the membrane and proteins degraded by Proteinase K. The samples were then subjected to phenol/chloroform extraction followed by ethanol precipitation. RNA was resuspended and ligated to 5′ RNA Linker (5'-GCUGAUGC UACGACCACAGGNNNU-3' 3'OH, 5'OH) with T4 RNA ligase at 16°C overnight. The RNA samples were then treated with DNase I followed by phenol/chloroform extraction and ethanol precipitation. The ligated RNA was reverse transcribed with P3 primer (5'-CAAGCAGAAGACGGCATACGAG-3') followed by PCR. PCR primers 5'-CAAGCAGAAGACGGCATACGAG and 5'-GCTGATGCTACGAC-CACAGG-3' were used. PCR product was analyzed on PAGE gel, bands corresponding to 75–150 nt were isolated and the DNA was extracted for sequencing.

52,415,296 total reads were obtained, among which 22,399,451 were mapped to the *C. elegans* genome (ce10). After removing PCR duplicates and non-unique reads, 688,276 unique reads were obtained. From these unique reads, overlapping CLIP tags were grouped to define CLIP peaks. We identified 1219 peaks distributed over 820 protein-coding genes, among which 410 genes are annotated with gene names (e.g. *scpl-1*), the remaining 410 genes being listed by sequence names (e.g. Y106G6D.8) (*Supplementary file 1*). Crosslinking induced mutation sites (CIMS) introduced by reverse transcriptase when bypassing the crosslink sites have been frequently detected in CLIP-seq data (*Ule et al., 2005*). Therefore we also used CIMS analysis method (*Zhang and Darnell, 2011*) when defining the peaks. We identified 1060 peaks located on 885 protein-coding genes (513 of which are functionally annotated) (*Supplementary file 1*). As expected, the two different peak calling methods (with or without CIMS) showed partial overlap, with 212 genes in common. We manually inspected all genomic loci of the functionally annotated genes on the two lists and identified 533 potential targets (*Supplementary file 1*).

## Nuclear CELF2 CLIP-seq in N2A cells

The Neuro2A cell line was obtained from ATCC (http://www.atcc.org/products/all/CCL-131.aspx#generalinformation) and cultured in DMEM (high glucose) containing 10% FBS with 5% $CO_2$. To generate an inducible Neuro2A stable cell line expressing biotin-tagged Celf2 (cDNA) that can be controlled to express at endogenous levels, we modified the original in vivo biotin-tagging system using a Tet-On retroviral inducible system from Clontech. Briefly, Celf2 was fused in-frame to the C terminus of the peptide MAGGLNDIFEAQKIEWHEDTGGGGSGGGGSGENLYFQSDYKDDDDK in the BLRP expression construct. Amino acids 1–20 represent a biotin ligase recognition peptide (BLRP), in which the lysine residue at position 13 is a substrate for the bacterial biotin ligase (BirA) upon co-expression in mammalian cells (*Liu et al., 2014*). The glycine-rich stretch following the BLRP sequence provides a spacer region, and the ENLYFQS sequence (in bold and underlined) provides a specific cleavage site for TEV protease (Life Technologies, Carlsbad, CA). Following the TEV site, there is a FLAG tag. The BLRP-Celf2 cassette was ligated into a retrovirus-based Tet-On vector then co-transfected with pCL-Ampho packaging plasmid into 293T cell line to prepare retrovirus. Retrovirus was then transduced into a parental Neuro2A stable line that was engineered to stably express

BirA and a Tet Repressor using a retroviral vector. G418 (150 µg/ml), hygromycin (200 µg/ml), and puromycin (0.7 µg/ml) were used for stable selection. BLRP-Celf2 cells were induced to express low level BLRP-Celf2 by doxycycline, then rinsed once with PBS, placed in HL-2000 Hybrilinker and irradiated at 150 mJ/cm² on ice for 1–1.5 min. Cells were washed twice with cold PBS and collected in PBS by scraping from culture flasks. Cells were gently resuspended in 500 µl 1x Hypotonic Buffer (500 µl per 10 cm dish or 10⁷ cells, scaled up as necessary) by pipetting up and down several times, followed by incubation on ice for 15 min. 25 µl detergent (10% $NP_40$) was added per 500 µl 1x Hypotonic Buffer followed by vortexing for 10 s at highest speed. The homogenate ws centrifuged for 10 min at 3,000 rpm at 4°C to get the nuclear fraction in the pellet. The pellet was then washed twice in 1x Hypotonic Buffer and resuspended in 1 ml CLIP lysis buffer (50 mM Tris–HCl, pH 7.4; 100 mM NaCl; 1 mM MgCl2; 0.1 mM CaCl2; 1% NP-40; 0.5% sodium deoxycholate; 0.1% SDS; Protease inhibitor and anti-RNase) per 100 µl pellet followed by sonication on ice. 5 ul Turbo DNase was added per 1 ml of lysates followed by incubation at 37°Cfor 3–5 min. Lysates were centrifuged at 16,000 g for 10 min at 4°C and supernatants collected and incubated with streptavidin magnetic beads (Sigma) overnight on a rotator.

Beads were then treated as for the CLIP-seq in *C. elegans* (see above) with some modification. Complexes were eluted from the beads and run on a native 10% Bis-Tris gel followed by transferring to a nitrocellulose membrane. The band corresponding to the CELF2/RNA complexes (65–100 kDa) was cut out of the membrane. After RNA extraction, RNA was resuspended and ligated to 5′ RNA Linker with T4 RNA ligase at 16°C overnight. The ligated RNA was reverse transcribed with P3 primer (5′-CAAGCAGAAGACGGCATACGAG-3′) followed by two rounds of PCR. PCR primers 5′-CAAGCAGAAGACGGCATACGAG and 5′-GCTGATGCTACGACCACAGG-3′ were used for the first round of PCR for 12 cycles. Primer 5′-CAAGCAGAAGACGGCATACGAG-3′ and barcoded PCR primers 5′- AATGATACGGCGACCACCGAGATNNNNGCTGATGCTACGACCACAGG-3′ were used for the second round of PCR for 4 cycles. PCR product was analyzed on PAGE gel, bands corresponding to 75–150 nt were isolated and the DNA was extracted for deep sequencing. We obtained 19,563,928 reads, among which 13,505,933 were mapped to mouse genome. After filtering PCR duplicates and non-unique tags, we obtained 4,587,451 unique tags. We used an algorithm in peak calling to include peaks with and without CIMS. The peaks with top 15% significance score were distributed over 2919 protein coding genes (*Supplementary file 4*).

## CLIP-seq bioinformatics analysis

The CLIP-seq data are available at the Gene Expression Omnibus under the accession number GSE78111.

### Read preprocessing

The quality of sequencing reads from fastq files was evaluated by FastQC (http://www.bioinformatics.babraham.ac.uk/projects/fastqc). Sequencing adapters and over-represented short sequences were trimmed using cutadapt (*Martin, 2011*).

### Read mapping

The trimmed reads from *unc-75* CLIP-seq or Celf2 CLIP-seq were mapped to the *C. elegans* genome (WS220/ce10) or *M. musculus* genome (GRCm38/mm10), respectively, by GSNAP(*Wu and Nacu, 2010*) (version 2015-09-29) with parameters of '-t 4 -k 10 -N 1'. The coordinates of mapped reads were evaluated and filtered by custom scripts. Specifically, reads with multiple coordinates and coordinates with duplicated reads were removed by using samtools (*Li et al., 2009*) and bedtools (*Quinlan and Hall, 2010*).

### Peak calling

Peak calling for CLIP-seq data was performed with and without considering CIMS (crosslinking-induced mutation site) as previously described (*Moore et al., 2014*; *Zhang and Darnell, 2011*), with some modifications. Using samtools and bedtools, mutation information in the read mapping was extracted, then CIMS analysis was applied to the read coordinates to detect significant peaks at FDR < 0.01 and m > 5.

## Motif finding

We used MEME-ChIP (*Machanick and Bailey, 2011*) for de novo identification of motifs enriched in CLIP-seq peaks with 20 bp extensions with both directions, using flanking sequences (100 bp length with 100 bp gap, extracted with bedtools) as background controls.

## RNA-seq

3'-RNA-seq was performed as previously described (*Fox-Walsh et al., 2011*). Briefly, total RNA was extracted from synchronized worms using Trizol (Invitrogen). cDNA synthesis was performed using SuperScript III (Invitrogen) and 3 µg of total RNA, together with 1 µl of 50 µM Biotin labeled oligodT and Adaptor-Random primers. cDNA was then purified using PCR purification kit (Macherey-Nagel) followed by terminal transferase treatment to add ddNTP to protect 3'-end. cDNA was then captured using streptavidin coated magnetic beads and primer extension was done on beads with Adaptor-Random primers. DNA was then eluted from beads by heating to 95°C before PCR amplification using barcoded primers. PCR products were then size selected and used for deep sequencing.

## Generation of *Celf2* floxed allele

Mouse husbandry and surgeries were performed under the supervision of the University of California San Diego Institutional Animal Care and Use Committee (IACUC). We used homologous recombination to create a 'floxed' *Celf2* allele consisting of *loxP* sites flanking *Celf2* exon 3 (*Figure 6—figure supplement 1*). A 1.1 kb genomic DNA fragment containing *Celf2* exon 3 was cloned into the targeting vector, flanked by two loxP sites. 2.2 kb genomic DNA upstream and a 4.8 kb genomic DNA fragment downstream to the 1.1 kb fragment were cloned into the targeting vector to generate 5' and 3' homology arms. The targeting vector was linearized by *Not* I digestion and transfected to CB6F1 ES cells. Homologous recombination was analyzed by Southern blotting using probes generated by PCR using the following primers: (5' probe: For-GGGACAGCAAGAAAGACAGT; Rev- CA TAGATGCAGCATTTAGTAGG. 3' probe: For-ACTCATTTCATTAAGGTTGTA; Rev-TAGTTTATCAG-GACCATTTG). Cells heterozygous for the targeted mutation were microinjected into blastocysts to obtain germ-line transmission following standard procedures. Mice were genotyped using PCR (primers: For-GAGGTGTCTGCCGAACT; Rev-CACTCAGTCCCTGTTTGTAA; Wt 470 bp, mutant 370 bp).

ZP3-Cre (*Lewandoski et al., 1997*) was used to generate *Celf2* null allele *Celf2⁻*. A *Parvalbumin-Cre* transgene (*Hippenmeyer et al., 2005*) was used to delete *Celf2* and *Rosa26-tdTomato* (*Madisen et al., 2010*) was used to label the *Cre⁺* cells. *Parvalbumin-Cre; Rosa26-tdTomato* mice were crossed to *Celf ⁺/⁻* mice to generate *PV-Cre; Rosa26-tdTomato; Celf2ⁿᵘˡˡ/⁺* progeny, which were then crossed to *Celf2ᶠˣ/ᶠˣ* mice to generate *PV-cre; Rosa26-tdTomato; Celf2ᶠˣ/⁻* animals for in vivo axon regeneration experiments.

## DRG explant and adult DRG culture

E13.5 mouse embryos were dissected from the uterus and put into cold F12 medium. The embryo was opened to expose the entire spinal cord which was then lifted to allow the dorsal root ganglia to be removed and transferred to culture medium (NB + 2% B27 + 10% FBS + L-glutamine) then to cover slips pre-coated with poly-ornithine and 5 µg/ml laminin. The DRGs were allowed to adhere for 4 hr before flooding the wells with culture medium. 24 hr later, DRGs were fixed with 4% PFA and stained with anti-Tuj1 (Covance, MMS-435P).

Adult (8 weeks) DRGs were dissected in cold F12 medium and then digested with 0.5 mg/ml collagenase (Roche, 10103578001) and 1 mg/ml dispase (Roche, 04942078001) for 40 min at 37°C followed by 0.125% trypsin digestion for 30 min at 37°C. Tissues were triturated in culture medium (NBA with 2% of B27 and 10% of FBS) with 1 ml tips and passed through a 0.45 µm cell strainer. Cells passing through the strainer were spun down and re-suspended in culture medium and plated to 12-well plates pre-coated with poly-ornithine and 5 µg/ml laminin. For in vitro axon regrowth analysis, 24 hr after plating, cells were re-suspended and re-plated to pre-coated cover slips. 24 hr later, re-plated cells were fixed and stained with anti-Tuj1.

## Sciatic nerve crush and axon regeneration analysis

2 month old mice were anesthetized with isoflurane and the sciatic nerve exposed by a small incision on the skin. The nerve was crushed with a pair of fine (#55) forceps for 20 s and the crush site marked using activated carbon powder (*Bauder and Ferguson, 2012*). 3 days later the mice were euthanized by $CO_2$ and sciatic nerves obtained for analysis. Sciatic nerves were fixed in 4% paraformaldehyde for 3 hr, then washed with PBS, immersed in 30% sucrose in PBS, cryopreserved in OCT compound (TissueTek) and cryosectioned at 10 μm thickness. Samples were immunostained with anti-SCG10 (1:3000) (Novus Biologicals STMN2 NBP1-49461). After staining, multiple images along the nerve were taken using a 10X objective (Zeiss LSM710) and montaged using Photoshop (Adobe). Representative images are shown in *Figure 6D*. We used Metamorph software to measure SCG10 staining fluorescence intensity in tdTomato positive axons (Cre-expressing neurons). At each distance point, at least 10–20 regions with tdTomato signal were randomly selected in the red channel and regions of interest (ROI) defined. These regions were then transferred to the green channel and the average intensity of green fluorescence measured. Average intensity from all the regions at each distance point was then normalized to the average intensity immediately proximal to the crush site.

## RNA extraction and RT-qPCR

Total RNA from worms or mouse tissues (E15.5 embryonic brains or dorsal root ganglia at different stages) were extracted using Trizol (Invitrogen) or RNeasy kit (Qiagen) following the manufacturers' protocols. First strand cDNA was reverse-transcribed using SuperScript III (Invitrogen). qPCR was run on Bio-Rad CFX96 Touch Real-Time PCR Detection System with iQ SYBR Supermix (Bio-Rad). Data were analyzed using CFX manager (Bio-Rad). PCR primers are listed in (*Supplementary file 6*).

## Statistical analysis

A two-tailed Student's test was used for comparisons of two groups. One-way ANOVA with Bonferroni post test was used to compare multiple groups in Prism (GraphPad, La Jolla, CA).

## Acknowledgements

We thank Michael Nonet and James Rand for reagents, and Edmund Hollis II for advice on sciatic nerve injury. We thank members of the Jin and Chisholm labs for discussion and advice. LC was partly supported by the Neuroplasticity of Aging Training Grant (NIH T32 AG000216). Some strains were obtained from the *Caenorhabditis* Genetics Center, which is supported by the NIH Office of Research Infrastructure Programs (P40 OD010440). LC was an Associate and YJ is an Investigator of the Howard Hughes Medical Institute. Supported by grants from NIH (HG004659 and GM049369 to XDF; R01NS093588, R01NS057317, and R56NS057317 to ADC and YJ).

## Additional information

### Funding

| Funder | Grant reference number | Author |
| --- | --- | --- |
| National Institutes of Health | HG004659 | Xiang-Dong Fu |
| National Institutes of Health | GM049369 | Xiang-Dong Fu |
| National Institute of Neurological Disorders and Stroke | R01NS093588 | Andrew D Chisholm Yishi Jin |
| National Institute of Neurological Disorders and Stroke | R01NS057317 | Andrew D Chisholm Yishi Jin |
| National Institute of Neurological Disorders and Stroke | R56NS057317 | Andrew D Chisholm Yishi Jin |
| Howard Hughes Medical Institute | Yishi Jin | Yishi Jin |

The funders had no role in study design, data collection and interpretation, or the decision to submit the work for publication.

## Author contributions
LC, Conception and design, Acquisition of data, Analysis and interpretation of data, Drafting or revising the article; ZL, Conception and design, Acquisition of data, Analysis and interpretation of data; BZ, YZ, Conception and design, Analysis and interpretation of data; CW, Conception and design, Contributed unpublished essential data or reagents; MGR, Contributed unpublished essential data or reagents; X-DF, Analysis and interpretation of data, Contributed unpublished essential data or reagents; ADC, YJ, Conception and design, Analysis and interpretation of data, Drafting or revising the article

## Author ORCIDs
Lizhen Chen, http://orcid.org/0000-0001-5313-7340
Bing Zhou, http://orcid.org/0000-0003-2846-1813
Andrew D Chisholm, http://orcid.org/0000-0001-5091-0537
Yishi Jin, http://orcid.org/0000-0002-9371-9860

## Ethics
Animal experimentation: This study was performed in strict accordance with the recommendations in the Guide for the Care and Use of Laboratory Animals of the National Institutes of Health. All of the animals were handled according to approved institutional animal care and use committee (IACUC) protocols (S13072) of the University of California. All surgery was performed under anesthesia, and every effort was made to minimize suffering.

# Additional files

### Supplementary files
• Supplementary file 1. mRNA targets identified by UNC-75 CLIP-seq in *C. elegans* neurons

• Supplementary file 2. Comparison of UNC-75 targets identified in this study with those defined by RNAseq.

• Supplementary file 3. Summary of expression patterns of mouse CELF transcripts based on Allen Brain Atlas.

• Supplementary file 4. mRNA targets identified by CELF2 CLIP-seq in N2a cells. The top 2919 CLIP-seq peaks are ranked by significance score.

• Supplementary file 5. *C. elegans* strains, transgenes, and clones.

• Supplementary file 6. Primers used for RT-qPCR analyses.

• Supplementary file 7. Raw measurement data of regrowth.

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
