## [Decision Letter]

Thank you for submitting your article "CELF RNA binding proteins promote axon regeneration in *C. elegans* and mammals through alternative splicing of syntaxins" for consideration by *eLife*. Your article has been reviewed by three peer reviewers, one of whom, Graeme W Davis, is a member of our Board of Reviewing Editors and the evaluation has been overseen by Senior Editor. The following individuals involved in review of your submission have agreed to reveal their identity: Aaron D Gitler (Reviewer #2).

The reviewers have discussed the reviews with one another and the Reviewing Editor has drafted this decision to help you prepare a revised submission.

Summary:

In this manuscript by Chen et al., the authors investigate the role of the RBP binding protein unc-75/CLEF and one of its targets unc-64/syntaxin in axonal regeneration. The authors provide compelling evidence that in *C. elegans* unc-75/CLEF is critical for regenerative axonal extension -but not for developmental axonal growth or regenerative growth cone formation, that it acts cell-autonomously, and that nuclear localization of unc-75/CLEF is critical for its pro-regenerative function. Through CLIP-seq the authors then identify neuronal unc-75/CLEF targets and demonstrate that unc-64/syntaxin is a major target critical for axonal regeneration.

The authors then complement their findings in *C. elegans* with definitive experiments in mice using conditional CLEF2 KO animals, a major step defining conserved gene function. Overall, this is an impressive manuscript that uncovers a conserved role for the unc-75/CLEF2 RNS binding protein in promoting axonal regeneration. Of particular note, this manuscript strongly reinforces the notion that genetic pathways critical for regeneration of single axons in *C. elegans* also function in multi- axonal regeneration of vertebrate peripheral nerves, a notion that no longer can be ignored.

This manuscript was reviewed by three individuals. All considered the data, interpretation or presentation to adhere to high standards and general interest consistent with publication at *eLife*. None of the reviewers had major concerns with the work. However, each had minor comments that the authors are asked to consider addressing prior to resubmission.

Minor Comments:

1) Please present non-normalized data in figure format or in table format to supplement the figures that contain normalized data.

2) Please re-visit the statistical analysis in Figure 6. The t-test on individual points is not the appropriate approach.

3) Are the quantifications of GFP intensity normalized somehow in Figure 3?

4) In Figure 6, it's stated that CELF2 is more effective than CELF4 at rescuing the unc-75 PLM defect. Are these mouse transgenes able to be expressed at similar levels?

5) Figure 7 shows axonal regeneration following crush injury in wild type and mutants, which is quantified in 7E. The quantification suggests an ~2fold reduction in relative intensity for most of the data points. Figure 7 however is suggestive of a much stronger phenotype. Were data from all 4 mutants pooled? Is there substantial variability from nerve to nerve?

6) The authors demonstrate that in *C. elegans* syntaxin promotes axonal regeneration. While not the major focus of this manuscript, a possible mechanism how syntaxin promotes regeneration is not provided. A discussion regarding possible mechanisms would be helpful.

7) The labeling of many figures could be improved for clarity. For example, the red asterisks indicating the cell bodies in Figure 1 are hardly visible. Similarly, the figure legends don't indicate what the red and orange arrows in 1D point at?

Reviewer #1:

The manuscript by Chen and colleagues places CEFL family RBPs in a new context, axonal regeneration. The topic should have a broad appeal; at a molecular level (RBP function) and phenomenologically (regeneration). The manuscript is well written and the data are nicely presented. In each instance, the effects being observed are large and convincing. The genetic analyses are thorough, including the generation of splicing reporters that are convincing and show large effect. The genetic data are appropriately interpreted. At the end of this manuscript, we are left with the finding that alternative syntaxin splicing is involved in the regenerative response, but there is no indication how syntaxin might participate. However, the paper, as written, is strong and the authors should not be required to make that next mechanistic leap in this paper. The story is solid and already makes the leap from work to mammalian systems, providing evidence of conserved signaling across distant species. I have only minor concerns.

Reviewer #2:

The authors have presented a thorough study that characterizes how CELF RNA binding proteins can regulate the alternative splicing of syntaxins, which may play a role in axon regeneration. The experiments had a logical flow starting from a genomics approach using CLIP-seq to identify RNA binding targets of UNC-75, followed by more functional genetic studies in nematodes and mice. Overall, the experiments were of high quality, the results are interesting, and will be of interest to the neuroscience community. I only have the following questions for the authors to consider.

1) Are the quantifications of GFP intensity normalized somehow in Figure 3?

2) In Figure 6, it's stated that CELF2 is more effective than CELF4 at rescuing the unc-75 PLM defect. Are these mouse transgenes able to be expressed at similar levels?

Reviewer #3

In this manuscript by Chen et al., the authors investigate the role of the RBP binding protein unc-75/CLEF and one of its targets unc-64/syntaxin in axonal regeneration. The authors provide compelling evidence that in *C. elegans* unc-75/CLEF is critical for regenerative axonal extension -but not for developmental axonal growth or regenerative growth cone formation, that it acts cell-autonomously, and that nuclear localization of unc-75/CLEF is critical for its pro-regenerative function. Through CLIP-seq the authors then identify neuronal unc-75/CLEF targets and demonstrate that unc-64/syntaxin is a major target critical for axonal regeneration.

What sets this manuscript apart from most is that the authors complement their findings in *C. elegans* with definitive experiments in mice using conditional CLEF2 KO animals! Overall, this is a very impressive manuscript that uncovers a conserved role for the unc-75/CLEF2 RNS binding protein in promoting axonal regeneration. I expect this manuscript to have a major impact on the field, in part because it delineates a novel, conserved pathway that promotes regeneration in vivo, but also because it strongly reinforces the notion that genetic pathways critical for regeneration of single axons in *C. elegans* also function in multi- axonal regeneration of vertebrate peripheral nerves, a notion that no longer can be ignored!

---

## [Author Response]

*1) Please present non-normalized data in figure format or in table format to supplement the figures that contain normalized data.*

We have provided a new [Supplementary-material SD7-data] for the non-normalized axon regrowth data.

Experimental variation in axon regrowth length results in part from known sensitivity of the laser to changes in temperature and humidity, and thus wild type controls show slight variation in mean regrowth between different days. To compare data sets from day to day, we routinely normalize axon regrowth to the average regrowth length of wild type controls performed on the same day as the experiment. Nevertheless in the data sets presented here such day-to-day variation was relatively low, so the normalized data display very similar pattern when compared to the non-normalized data.

2) Please re-visit the statistical analysis in Figure 6. The t-test on individual points is not the appropriate approach.

This point might be some kind of misunderstanding. As stated in the original MS, the analysis in Figure 6 uses one way ANOVA, not the t-test. Nonetheless, we have also rechecked the statistical analyses in other related figures.

3) Are the quantifications of GFP intensity normalized somehow in Figure 3?

The GFP intensity in Figure 3 is not normalized. The bar graph shows the mean intensity ± SEM. All genotypes were imaged using the same microscope settings with same laser power. To keep laser illumination uniform across genotypes we used a relatively high laser power. This may potentially underestimate the intensity of bright GFP signals due to saturation. Thus the reduction in UNC-64A levels in unc-75 may be greater than estimated.

4) In Figure 6, it's stated that CELF2 is more effective than CELF4 at rescuing the unc-75 PLM defect. Are these mouse transgenes able to be expressed at similar levels?

In the transgenes used here (*juEx5488* and *juEx5490*) CELF2 and CELF4 were tagged with mCherry, and by visual inspection the fusion proteins were expressed at similar levels (additional transgenic lines were tested, and representative data are shown). We have added a statement to this effect in the legend to Figure 6. As this is a minor control we have elected to mention this as data not shown, although if necessary the data can be made available for review.

5) Figure 7 shows axonal regeneration following crush injury in wild type and mutants, which is quantified in 7E. The quantification suggests an ~2fold reduction in relative intensity for most of the data points. Figure 7 however is suggestive of a much stronger phenotype. Were data from all 4 mutants pooled? Is there substantial variability from nerve to nerve?

The graph (assuming 7D/E is a typo for 6D/E) shows mean intensity ± SEM, with data pooled from 4 mutants and 5 controls. There was some variability from nerve to nerve in both controls and mutants. We have selected the images that are representative of the mean intensity to convey the mutant phenotypes

*6) The authors demonstrate that in C. elegans syntaxin promotes axonal regeneration. While not the major focus of this manuscript, a possible mechanism how syntaxin promotes regeneration is not provided. A discussion regarding possible mechanisms would be helpful.*

We thank the reviewer for the comment and have added a paragraph to the Discussion surveying previous relevant findings on syntaxins and axon growth:

“The mechanism by which syntaxin contributes to regenerative axonal elongation remains to be elucidated. Because unc-64 mutants are defective in axon extension rather than growth cone reformation, we believe it is less likely that syntaxin is required for membrane resealing immediately after injury. […] Finally, syntaxin might contribute to membrane expansion via a non-fusogenic mechanism as shown for the SNARE Sec22b (Petkovic et al., 2014).”

7) The labeling of many figures could be improved for clarity. For example, the red asterisks indicating the cell bodies in Figure 1 are hardly visible. Similarly, the figure legends don't indicate what the red and orange arrows in 1D point at?

We appreciate the comments and have improved the labeling and figure legends.